# GPU-Accelerated Counterfactual Regret Minimization

## Abstract

Counterfactual regret minimization is a family of algorithms of no-regret learning dynamics capable of solving large-scale imperfect information games. We propose implementing this algorithm as a series of dense and sparse matrix and vector operations, thereby making it highly parallelizable for a graphical processing unit, at a cost of higher memory usage. Our experiments show that our implementation performs up to about 401.2 times faster than OpenSpiel's Python implementation and, on an expanded set of games, up to about 203.6 times faster than OpenSpiel's C++ implementation and the speedup becomes more pronounced as the size of the game being solved grows.

## 1 Introduction

Counterfactual regret minimization (CFR) (Zinkevich et al., 2007) and its variants dominated the development of AI agents for large imperfect information games like *Poker* (Tammelin et al., 2015; Moravčík et al., 2017; Brown & Sandholm, 2018; 2019b) and *The Resistance: Avalon* (Serrino et al., 2019) and were components of ReBeL (Brown et al., 2020) and student of games (Schmid et al., 2023). Notable variants of CFR are as follows: CFR+ by Tammelin (2014) (optionally) eliminates the averaging step while improving the convergence rate; Sampling variants (Lanctot et al., 2009) makes a complete recursive tree traversal unnecessary; Burch et al. (2014) proposes CFR-D in which games are decomposed into subgames; Brown & Sandholm (2019a) explores modifying CFR such as alternate weighted averaging (and discounting) schemes; Xu et al. (2024) learns a discounting technique from smaller games to be used in larger games.

We propose implementing CFR as a series of linear algebra operations, as done by Graph-BLAS (Kepner et al., 2016) for graph algorithms, thereby parallelizing it for a graphical processing unit (GPU) at a cost of higher memory usage. We analyze the runtimes of our implementation with both computer processing unit (CPU) and GPU backends and compare them to Google DeepMind's OpenSpiel (Lanctot et al., 2020) implementations in Python and C++ on 20 games of differing sizes.

Our experiments show that, compared to Google DeepMind OpenSpiel's (Lanctot et al., 2020) Python implementation, our GPU implementation performs about 3.5 times slower for small games but is up to about 401.2 times faster for large games. Against their C++ implementation, our performance with a GPU is up to about 85.5 times slower for small games, but, on an expanded set of games, is up to about 203.6 times faster for large games. Even without a GPU, our implementation shows speedups compared to the OpenSpiel baselines (from about 1.5 to 46.8 times faster than their Python implementation and from 16.8 times slower to 4.5 times faster than theirs in C++). We see that the speedup becomes more pronounced as the size of the game solved grows.

## 2 Background

The background of our work and the notations we use throughout this paper is introduced below.

### 2.1 Finite Extensive-Form Games

An extensive-form game is a representation of games that allow the specification of the rules of the game, information sets (infosets), actions, actors (players and the nature), chances, and payoffs.

**Definition 1** The formal definition of a **finite extensive-form game** (Osborne & Rubinstein, 1994) is a structure $\mathcal{G} = \langle \mathcal{T}, \mathbb{H}, f_h, \mathbb{A}, f_a, \mathbb{I}, f_i, \sigma_0, u \rangle$ where:

- $\mathcal{T} = \langle \mathbb{V}, v_0, \mathbb{T}, f_{Pa} \rangle$ is a **finite game tree** with a **finite set of nodes** (i.e., vertices) $\mathbb{V}$, a unique **initial node** (i.e., a root) $v_0 \in \mathbb{V}$, a **finite set of terminal nodes** (i.e., leaves) $\mathbb{T} \subseteq \mathbb{V}$, and a **parent function** $f_{Pa} : \mathbb{V}_+ \to \mathbb{D}$ that maps a non-initial node (i.e., a non-root) $v_+ \in \mathbb{V}_+$ to an immediate predecessor (i.e., a parent) $d \in \mathbb{D}$, with $\mathbb{V}_+ = \mathbb{V}\backslash\{v_0\}$ the finite set of non-initial nodes (i.e., non-roots) and $\mathbb{D} = \mathbb{V}\backslash\mathbb{T}$ the finite set of decision nodes (i.e., internal vertices),

- $\mathbb{H}$ is a **finite set of infosets**, $f_h : \mathbb{D} \to \mathbb{H}$ is an **information partition** of $\mathbb{D}$ associating each decision node $d \in \mathbb{D}$ to an infoset $h \in \mathbb{H}$,

- $\mathbb{A}$ is a **finite set of actions**, $f_a : \mathbb{V}_+ \to \mathbb{A}$ is an **action partition** of $\mathbb{V}_+$ associating each non-initial node $v_+ \in \mathbb{V}_+$ to an action $a \in \mathbb{A}$ such that $\forall d \in \mathbb{D}$ the restriction $f_{a,d} : S(d) \to A(f_h(d))$ is a bijection, with $S(d \in \mathbb{D}) = \{v_+ \in \mathbb{V}_+ : f_{Pa}(v_+) = d\}$ the finite set of immediate successors (i.e., children) of a node $d \in \mathbb{D}$ and $A(h \in \mathbb{H}) = \{a \in \mathbb{A} : [\exists v_+ \in \mathbb{V}_+](f_h(f_{Pa}(v_+)) = h \land f_a(v_+) = a)\}$ the finite set of available actions at an infoset $h \in \mathbb{H}$,

- $\mathbb{I}$ is a **finite set of (rational) players and, optionally, the nature** (i.e., chance) $i_0 \in \mathbb{I}$, $f_i : \mathbb{H} \to \mathbb{I}$ is a **player partition** of $\mathbb{H}$ associating each infoset $h \in \mathbb{H}$ to a player $i \in \mathbb{I}$,

- $\sigma_0 : \mathbb{Q}_0 \to [0,1]$ is a **chance probabilities function** that associates each pair of a nature infoset and an available action $(h_0, a) \in \mathbb{Q}_0$ to an independent probability value, with $\mathbb{Q}_j = \{(h,a) \in \mathbb{Q} : h \in \mathbb{H}_j\}$ the finite set of pairs of a player infoset $h_j \in \mathbb{H}_j$ and an available action $a \in A(h_j)$, $\mathbb{Q} = \{(h,a) \in \mathbb{H} \times \mathbb{A} : a \in A(h)\}$ the finite set of pairs of an infoset $h \in \mathbb{H}$ and an available action $a \in A(h)$, and $\mathbb{H}_j = \{h \in \mathbb{H} : f_i(h) = i_j\}$ the finite set of infosets associated with a player $i_j \in \mathbb{I}$, and

- $u : \mathbb{T} \times \mathbb{I}_+ \to \mathbb{R}$ is a **utility function** that associates each pair of a terminal node $t \in \mathbb{T}$ and a (rational) player $i_+ \in \mathbb{I}_+$ to a real payoff value. $\mathbb{I}_+ = \mathbb{I}\backslash\{i_0\}$ is the finite set of (rational) players.

## 2.2 Nash Equilibrium

Each player $i_j \in \mathbb{I}$ selects a **player strategy** $\sigma_j : \mathbb{Q}_j \to [0,1]$ from a **set of player strategies** $\Sigma_j$. A player strategy $\sigma_j \in \Sigma_j$ associates, for each player infoset $h_j \in \mathbb{H}_j$, a probability distribution over a finite set of available actions $A(h_j)$. A **strategy profile** $\sigma : \mathbb{Q} \to [0,1]$ is a direct sum of the strategies of each player $\sigma = \bigoplus_{i_j \in \mathbb{I}} \sigma_j$ which, for each infoset $h \in \mathbb{H}$, gives a probability distribution over a finite set of available actions $A(h)$. $\Sigma$ is a set of strategy profiles. $\sigma_{-j} = \bigoplus_{i_k \in \mathbb{I}\backslash\{i_j\}} \sigma_k$ is a direct sum of all player strategies in $\sigma$ except $\sigma_j$ (i.e., that of player $i_j \in \mathbb{I}$).

Let $\pi : \Sigma \times \mathbb{V} \to \mathbb{R}$ be a probability of reaching a vertex $v \in \mathbb{V}$ following a strategy profile $\sigma \in \Sigma$.

$$\pi(\sigma \in \Sigma, v \in \mathbb{V}) = \begin{cases} \sigma(f_h(f_{Pa}(v)), f_a(v))\pi(\sigma, f_{Pa}(v)) & v \in \mathbb{V}_+ \\ 1 & v = v_0 \end{cases}$$

Then, define $\hat{u} : \Sigma \times \mathbb{I} \to \mathbb{R}$ to be an expected payoff of a (rational) player $i_+ \in \mathbb{I}_+$, following a strategy profile $\sigma \in \Sigma$.

$$\hat{u}(\sigma \in \Sigma, i_+ \in \mathbb{I}_+) = \sum_{t \in \mathbb{T}} \pi(\sigma, t)u(t, i_+)$$

A strategy profile $\sigma^* \in \Sigma$ is a **Nash equilibrium**, a traditional solution concept for non-cooperative games, if no player stands to gain by deviating from the strategy profile.

$$\forall i_{+,j} \in \mathbb{I}_+ \quad \hat{u}(\sigma^*, i_{+,j}) \geqslant \max_{\sigma'_j \in \Sigma_j} \hat{u}(\sigma'_j \oplus \sigma^*_{-j}, i_{+,j})$$

A strategy profile that approximates a Nash equilibrium $\sigma^*$ is an $\epsilon$-**Nash equilibrium** $\sigma^{*,\epsilon} \in \Sigma$ if

$$\forall i_{+,j} \in \mathbb{I}_+ \quad \hat{u}(\sigma^{*,\epsilon}, i_{+,j}) + \epsilon \geqslant \max_{\sigma'_j \in \Sigma_j} \hat{u}(\sigma'_j \oplus \sigma^{*,\epsilon}_{-j}, i_{+,j})$$

### 2.3 COUNTERFACTUAL REGRET MINIMIZATION

Define $\check{u} : \Sigma \times \mathbb{V} \times \mathbb{I}_+ \to \mathbb{R}$ as an expected payoff of a (rational) player $i_+ \in \mathbb{I}_+$ at a node $v \in \mathbb{V}$, following a strategy profile $\sigma \in \Sigma$.

$$\check{u}(\sigma \in \Sigma, v \in \mathbb{V}, i_+ \in \mathbb{I}_+) = \begin{cases} \sum_{s \in S(v)} \sigma(f_h(v), f_a(s))\check{u}(\sigma, s, i_+) & v \in \mathbb{D} \\ u(v, i_+) & v \in \mathbb{T} \end{cases} \tag{1}$$

Let $\check{\pi} : \Sigma \times \mathbb{V} \times \mathbb{I} \to \mathbb{R}$ be a probability of reaching a vertex $v \in \mathbb{V}$ following a strategy profile $\sigma \in \Sigma$ while ignoring a strategy of a player $i \in \mathbb{I}$.

$$\check{\pi}(\sigma \in \Sigma, v \in \mathbb{V}, i \in \mathbb{I}) = \begin{cases} \check{\pi}(\sigma, f_{Pa}(v), i) \begin{cases} \sigma(f_h(f_{Pa}(v)), f_a(v)) & f_i(f_h(f_{Pa}(v))) \neq i \\ 1 & f_i(f_h(f_{Pa}(v))) = i \end{cases} & v \in \mathbb{V}_+ \\ 1 & v = v_0 \end{cases} \tag{2}$$

The below definition shows a counterfactual reach probability $\tilde{\pi} : \Sigma \times \mathbb{H} \to \mathbb{R}$.

$$\tilde{\pi}(\sigma \in \Sigma, h \in \mathbb{H}) = \sum_{d \in \mathbb{D}: f_h(d)=h} \check{\pi}(\sigma, d, f_i(h)) \tag{3}$$

Let $\bar{\pi} : \Sigma \times \mathbb{H} \to \mathbb{R}$ be "player" reach probability, with $\hat{\pi} : \Sigma \times \mathbb{V} \times \mathbb{I} \to \mathbb{R}$ the probability of reaching a vertex $v \in \mathbb{V}$, only considering the strategy of one particular player.

$$\hat{\pi}(\sigma \in \Sigma, v \in \mathbb{V}, i \in \mathbb{I}) = \begin{cases} \check{\pi}(\sigma, f_{Pa}(v), i) \begin{cases} \sigma(f_h(f_{Pa}(v)), f_a(v)) & f_i(f_h(f_{Pa}(v))) = i \\ 1 & f_i(f_h(f_{Pa}(v))) \neq i \end{cases} & v \in \mathbb{V}_+ \\ 1 & v = v_0 \end{cases} \tag{4}$$

$$\bar{\pi}(\sigma \in \Sigma, h \in \mathbb{H}) = \sum_{d \in \mathbb{D}: f_h(d)=h} \hat{\pi}(\sigma, d, f_i(h)) \tag{5}$$

Now, let $\tilde{u} : \Sigma \times \mathbb{H}_+ \to \mathbb{R}$ be a counterfactual utility, with $\mathbb{H}_+ = \mathbb{H} \backslash \mathbb{H}_0$ the finite set of infosets associated with (rational) players.

$$\tilde{u}(\sigma \in \Sigma, h_+ \in \mathbb{H}_+) = \frac{\sum_{d \in \mathbb{D}: f_h(d)=h_+} \check{\pi}(\sigma, d, f_i(h_+))\check{u}(\sigma, d, f_i(h_+))}{\tilde{\pi}(\sigma, h_+)} \tag{6}$$

$\sigma|_{h \to a} \in \Sigma$ is an overridden strategy profile of $\sigma$ where an action $a \in A(h)$ is always taken at an infoset $h \in \mathbb{H}$.

$$\sigma|_{h \to a}((h', a') \in \mathbb{Q}) = \begin{cases} \mathbf{1}_{a=a'} & h = h' \\ \sigma(h', a') & h \neq h' \end{cases}$$

$\tilde{r} : \Sigma \times \mathbb{Q}_+ \to \mathbb{R}$ is the instantaneous counterfactual regret, with $\mathbb{Q}_+ = \mathbb{Q} \backslash \mathbb{Q}_0$ the finite set of pairs of a (rational) player infoset $h_+ \in \mathbb{H}_+$ and an available action $a \in A(h_+)$.

$$\tilde{r}(\sigma \in \Sigma, (h_+, a) \in \mathbb{Q}_+) = \tilde{\pi}(\sigma, h_+)(\tilde{u}(\sigma|_{h_+ \to a}, h_+) - \tilde{u}(\sigma, h_+)) \tag{7}$$

$\bar{r}^{(T)} : \mathbb{Q}_+ \to \mathbb{R}$ is the average counterfactual regret at an iteration $T$. $\sigma^{(\tau)} \in \Sigma$ is the strategy at $\tau$.

$$\bar{r}^{(T)}(q_+ \in \mathbb{Q}_+) = \frac{1}{T} \sum_{\tau=1}^{T} \tilde{r}(\sigma^{(\tau)}, q_+) \tag{8}$$

The strategy profile for Iteration $T + 1$ is $\sigma^{(T+1)} \in \Sigma$.

$$\sigma^{(T+1)}((h,a) \in \mathbb{Q}) = \begin{cases} \begin{cases} \frac{(\bar{r}^{(T)}(h,a))^+}{\sum_{a' \in A(h)} (\bar{r}^{(T)}(h,a'))^+} & \sum_{a' \in A(h)} (\bar{r}^{(T)}(h,a'))^+ > 0 \\ \frac{1}{|A(h)|} & \sum_{a' \in A(h)} (\bar{r}^{(T)}(h,a'))^+ = 0 \end{cases} & (h,a) \in \mathbb{Q}_+ \\ \sigma_0(h,a) & (h,a) \in \mathbb{Q}_0 \end{cases} \tag{9}$$

**CFR** (Zinkevich et al., 2007) is an algorithm that iteratively approximates a coarse correlated equilibrium $\bar{\sigma}^{(T)} : \mathbb{Q} \to \mathbb{R}$ (Hart & Mas-Colell, 2000).

$$\bar{\sigma}^{(T)}((h,a) \in \mathbb{Q}) = \frac{\sum_{\tau=1}^{T} \bar{\pi}(\sigma^{(\tau)}, h) \sigma^{(\tau)}(h,a)}{\sum_{\tau=1}^{T} \bar{\pi}(\sigma^{(\tau)}, h)} \tag{10}$$

Define $r^{(T)} : \mathbb{I}_+ \to \mathbb{R}$ as the average overall regret of a (rational) player $i_{+,j} \in \mathbb{I}_+$ at an iteration $T$.

$$r^{(T)}(i_{+,j} \in \mathbb{I}_+) = \frac{1}{T} \max_{\sigma'_j \in \Sigma_j} \sum_{\tau=1}^{T} (\hat{u}(\sigma'_j \oplus \sigma^{(\tau)}_{-j}, i_{+,j}) - \hat{u}(\sigma^{(\tau)}, i_{+,j}))$$

In 2-player zero-sum games, if $\forall i_+ \in \mathbb{I}_+ \ r^{(T)}(i_+) \leqslant \epsilon$, the average strategy $\bar{\sigma}^{(T)}$ (at an iteration $T$) is also a $2\epsilon$-Nash equilibrium $\sigma^{*,2\epsilon} \in \Sigma$ (Zinkevich et al., 2007).

## 2.4 PRIOR USAGES OF GPUs FOR CFR

In the mainstream literature, algorithms inspired by CFR or using CFR as a subcomponent like DeepStack (Moravčík et al., 2017), Student of Games (Schmid et al., 2023), and ReBeL (Brown et al., 2020) only perform a limited lookahead instead of a complete game tree traversal. A neural network-based value function is typically used to evaluate the heuristic value of a node – GPUs can be utilized for the evaluation of these networks. Besides the fact that the vanilla CFR considers the entire game tree and does not use a value function, our approach differs significantly in that we use the GPU to parallelize CFR at every step of the process.

A number of obscure unpublished works (Reis, 2015; Rudolf, 2021) have implemented CFR directly on CUDA and found orders of magnitude improvements in performance. However, in the implementation by Rudolf (2021), every thread assigned to each node moves up the game tree (toward the root), thus resulting in a quadratic number of visits to the game tree per iteration in the worst case. The implementation by Reis (2015) is superior in that only one visit is made at each node per iteration by doing level-by-level updates (an approach we also use). However, aside from reproducibility issues with his work[1], both require each thread to perform a "large number of control flow statements" – a limitation mentioned by Reis (2015) – and require more generalized kernel instructions.

Our approach addresses these issues by framing this problem as a series of linear algebra operations, and the utilization of GPUs for this task is an extremely well-studied problem in the field of systems, and can take advantage of optimized opcodes for these operations. Our implementation is also compatible with discrete games in OpenSpiel, which are commonly used as benchmarks for evaluating newly proposed CFR variants, unlike the work by Reis (2015) whose compatible games are limited to customized poker variants. In addition, our pure Python code is open-source.

---

[1]Reis's thesis contains screenshots of his code as figures that cannot compile due to syntax errors. For example, we point out the missing semicolon in Line 4 of Figure 12 and the mismatched square brace in Line 8 of Figure 18. Aside from the obvious errors, the thesis's code snippets do not handle chance nodes, decision nodes, and terminal nodes separately.

## 3 IMPLEMENTATION

In order to highly parallelize the execution of CFR, we implement the algorithm as a series of dense and sparse matrix and vector operations and avoid recursive game tree traversals.

### 3.1 SETUP

Calculating expected payoffs of players $\tilde{u} : \Sigma \times \mathbb{V} \times \mathbb{I}_+ \to \mathbb{R}$ in Equation 1, and reach probabilities $\tilde{\pi} : \Sigma \times \mathbb{V} \times \mathbb{I} \to \mathbb{R}$ in Equation 2 and $\hat{\pi} : \Sigma \times \mathbb{V} \times \mathbb{I} \to \mathbb{R}$ in Equation 5 are problems of dynamic programming on trees. To calculate these values with linear algebra operations, we represent the game tree $\mathcal{T}$ as an adjacency matrix $\boldsymbol{G} \in \mathbb{R}^{\mathbb{V}^2}$ and the level graphs of the game tree $\mathcal{T}$ as adjacency matrices $\boldsymbol{L}^{(1)}, \boldsymbol{L}^{(2)}, \ldots, \boldsymbol{L}^{(D)} \in \mathbb{R}^{\mathbb{V}^2}$, with $D = \max_{t \in \mathbb{T}} d_{\mathcal{T}}(t)$ the maximum depth of any node in the game tree $\mathcal{T}$ and $d_{\mathcal{T}} : \mathbb{V} \to \mathbb{Z}$ the depth of a vertex $v \in \mathbb{V}$ in the game tree $\mathcal{T}$ from the root $v_0$.

$$\boldsymbol{G} = \left( \begin{cases} \mathbf{1}_{v = f_{Pa}(v')} & v \in \mathbb{D} \wedge v' \in \mathbb{V}_+ \\ 0 & v \in \mathbb{T} \vee v' = v_0 \end{cases} \right)_{(v,v') \in \mathbb{V}^2}$$

$$\forall l \in [1, D] \cap \mathbb{Z} \quad \boldsymbol{L}^{(l)} = \left( \begin{cases} \mathbf{1}_{v = f_{Pa}(v') \wedge d_{\mathcal{T}}(v') = l} & v \in \mathbb{D} \wedge v' \in \mathbb{V}_+ \\ 0 & v \in \mathbb{T} \vee v' = v_0 \end{cases} \right)_{(v,v') \in \mathbb{V}^2}$$

$\boldsymbol{M}^{(Q_+,V)} \in \mathbb{R}^{\mathbb{Q}_+ \times \mathbb{V}}, \boldsymbol{M}^{(H_+,Q_+)} \in \mathbb{R}^{\mathbb{H}_+ \times \mathbb{Q}_+}, \boldsymbol{M}^{(V,I_+)} \in \mathbb{R}^{\mathbb{V} \times \mathbb{I}_+}$ are masking matrices that represent the game $\mathcal{G}$. Matrix $\boldsymbol{M}^{(Q_+,V)}$ describes whether a node $v \in \mathbb{V}$ is a result of an action from a (rational) player infoset $(h_+, a) \in \mathbb{Q}_+$. Matrix $\boldsymbol{M}^{(H_+,Q_+)}$ describes whether a (rational) player infoset $h_+ \in \mathbb{H}_+$ is the first element of the corresponding (rational) player infoset-action pair $(h_+, a) \in \mathbb{Q}_+$. Finally, matrix $\boldsymbol{M}^{(V,I_+)}$ describes whether a node $v \in \mathbb{V}$ has a parent whose associated infoset's associated player is $i_+ \in \mathbb{I}_+$ (i.e., which player $i_+ \in \mathbb{I}_+$ acted to reach a node $v \in \mathbb{V}$). Note that we omit the nature player $i_0$ and related infosets $\mathbb{H}_0$ and infoset-action pairs $\mathbb{Q}_0$ as only the strategies of (rational) players are updated by the algorithm. These mask-like matrices are later used to "select" the values associated with a player, action, node, or infoset during the iteration.

$$\boldsymbol{M}^{(Q_+,V)} = \left( \begin{cases} \mathbf{1}_{q_+ = (f_h(f_{Pa}(v)), f_a(v))} & v \in \mathbb{V}_+ \\ 0 & v = v_0 \end{cases} \right)_{(q_+, v) \in \mathbb{Q}_+ \times \mathbb{V}}$$

$$\boldsymbol{M}^{(H_+,Q_+)} = \left( \mathbf{1}_{h_+ = h'_+} \right)_{(h_+, (h'_+, a)) \in \mathbb{H}_+ \times \mathbb{Q}_+} \quad \boldsymbol{M}^{(V,I_+)} = \left( \begin{cases} \mathbf{1}_{f_i(f_h(f_{Pa}(v))) = i_+} & v \in \mathbb{V}_+ \\ 0 & v = v_0 \end{cases} \right)_{(v, i_+) \in \mathbb{V} \times \mathbb{I}_+}$$

$\boldsymbol{G}, \boldsymbol{L}^{(1)}, \boldsymbol{L}^{(2)}, \ldots, \boldsymbol{L}^{(D)}, \boldsymbol{M}^{(Q_+,V)}, \boldsymbol{M}^{(H_+,Q_+)}, \boldsymbol{M}^{(V,I_+)}$ are constant matrices. In the games we experiment on, all aforesaid matrices except $\boldsymbol{M}^{(V,I_+)}$ are highly sparse (as demonstrated in Appendix C).[2] As such, they are implemented as sparse matrices in a compressed sparse row (CSR) format. Matrix $\boldsymbol{M}^{(V,I_+)}$ and all other defined matrices and vectors are dense.

Define a vector $\boldsymbol{s}^{(\sigma_0)}$ representing the probabilities of nature infoset-action pairs $\mathbb{Q}_0$.

$$\boldsymbol{s}^{(\sigma_0)} = \left( \begin{cases} \begin{cases} \sigma_0(f_h(f_{Pa}(v)), f_a(v)) & f_h(f_{Pa}(v)) \in \mathbb{H}_0 \\ 0 & f_h(f_{Pa}(v)) \in \mathbb{H}_+ \end{cases} & v \in \mathbb{V}_+ \\ 0 & v = v_0 \end{cases} \right)_{v \in \mathbb{V}}$$

$\boldsymbol{\sigma} \in \mathbb{R}^{\mathbb{Q}_+}$ is the strategy over (rational) player infoset-action pairs $\mathbb{Q}_+$ at an iteration $T$.

$$\boldsymbol{\sigma} = \left( \sigma^{(T)}(q_+) \right)_{q_+ \in \mathbb{Q}_+}$$

---

[2]The sparsity of $\boldsymbol{M}^{(V,I_+)}$ depends on the number of (rational) players. For games with many players, it may be more efficient to implement this as sparse as well.

A vector $\boldsymbol{\sigma}^{(T=1)} \in \mathbb{R}^{\mathbb{Q}_+}$ representing the initial strategy profile (i.e., at $T = 1$) is shown below.

$$\boldsymbol{\sigma}^{(T=1)} = \left(\sigma^{(1)}(q_+)\right)_{q_+ \in \mathbb{Q}_+} = \left(\frac{1}{|A(h_+)|}\right)_{(h_+,a) \in \mathbb{Q}_+} = \mathbf{1}_{|\mathbb{Q}_+|} \oslash \left(\left(\boldsymbol{M}^{(H_+,Q_+)}\right)^\top \left(\left(\boldsymbol{M}^{(H_+,Q_+)}\right) \mathbf{1}_{|\mathbb{Q}_+|}\right)\right)$$

On each iteration, the strategy at the next iteration $\boldsymbol{\sigma}' = \left(\sigma^{(T+1)}(q_+)\right)_{q_+ \in \mathbb{Q}_+}$ is calculated using $\boldsymbol{\sigma}$.

## 3.2 ITERATION

### 3.2.1 TREE TRAVERSAL

Let a vector $\boldsymbol{s} \in \mathbb{R}^{\mathbb{V}}$ represent the probabilities of taking an action that reaches a node $v \in \mathbb{V}$ at an iteration $T$. This value is irrelevant for the unique initial node $v_0$.

$$\boldsymbol{s} = \left(\begin{cases} \sigma^{(T)}(f_h(f_{Pa}(v)), f_a(v)) & v \in \mathbb{V}_+ \\ 0 & v = v_0 \end{cases}\right)_{v \in \mathbb{V}} = \left(\boldsymbol{M}^{(Q_+,V)}\right)^\top \boldsymbol{\sigma} + \boldsymbol{s}^{(\sigma_0)}$$

For later use, we also broadcast the vector $\boldsymbol{s}$ to be a matrix $\boldsymbol{S} \in \mathbb{R}^{\mathbb{V}^2}$. This is defined only for notational convenience and, in our implementation, this matrix is not actually stored in memory.

$$\boldsymbol{S} = (\boldsymbol{s}_{v'})_{(v,v') \in \mathbb{V}^2}$$

The recurrence relations of the expected payoffs of (rational) players $\check{u} : \Sigma \times \mathbb{V} \times \mathbb{I}_+ \to \mathbb{R}$ (see Equation 1) is expressed with matrices. Define the following matrices $\check{\boldsymbol{U}}^{(1)}, \check{\boldsymbol{U}}^{(2)}, \ldots, \check{\boldsymbol{U}}^{(D+1)} \in \mathbb{R}^{\mathbb{V} \times \mathbb{I}_+}$:

$$\forall l \in [1, D+1] \cap \mathbb{Z} \quad \check{\boldsymbol{U}}^{(l)} = \left(\begin{cases} \check{u}(\sigma^{(T)}, v, i_+) & d_{\mathcal{T}}(v) \geqslant l-1 \vee v \in \mathbb{T} \\ 0 & d_{\mathcal{T}}(v) < l-1 \wedge v \in \mathbb{D} \end{cases}\right)_{(v,i_+) \in \mathbb{V} \times \mathbb{I}_+}$$

$$\check{\boldsymbol{U}}^{(D+1)} = \left(\begin{cases} u(v, i_+) & v \in \mathbb{T} \\ 0 & v \in \mathbb{D} \end{cases}\right)_{(v,i_+) \in \mathbb{V} \times \mathbb{I}_+} \quad \forall l \in [1, D] \cap \mathbb{Z} \quad \check{\boldsymbol{U}}^{(l)} = \left(\boldsymbol{L}^{(l)} \odot \boldsymbol{S}\right) \check{\boldsymbol{U}}^{(l+1)} + \check{\boldsymbol{U}}^{(l+1)} \tag{11}$$

Let $\check{\boldsymbol{U}} \in \mathbb{R}^{\mathbb{V} \times \mathbb{I}_+}$ represent $\check{u} : \Sigma \times \mathbb{V} \times \mathbb{I}_+ \to \mathbb{R}$.

$$\check{\boldsymbol{U}} = \left(\check{u}(\sigma^{(T)}, v, i_+)\right)_{(v,i_+) \in \mathbb{V} \times \mathbb{I}_+} = \check{\boldsymbol{U}}^{(1)}$$

Let $\check{\boldsymbol{S}} \in \mathbb{R}^{\mathbb{V} \times \mathbb{I}_+}$ be a matrix to be used in a later calculation.

$$\check{\boldsymbol{S}} = \left(\begin{cases} \boldsymbol{s}_v & \left(\boldsymbol{M}^{(V,I_+)}\right)_{v,i_+} = 0 \\ 1 & \left(\boldsymbol{M}^{(V,I_+)}\right)_{v,i_+} = 1 \end{cases}\right)_{(v,i_+) \in \mathbb{V} \times \mathbb{I}_+} \tag{12}$$

In order to represent a restriction (ignoring nature) of the "excepted" reach probabilities (defined in Equation 2) $\check{\pi} : \Sigma \times \mathbb{V} \times \mathbb{I} \to \mathbb{R}$ with matrices, we, again, express the recurrence relations with matrices. We therefore define the following matrices: $\check{\boldsymbol{\Pi}}^{(0)}, \check{\boldsymbol{\Pi}}^{(1)}, \check{\boldsymbol{\Pi}}^{(2)}, \ldots, \check{\boldsymbol{\Pi}}^{(D)} \in \mathbb{R}^{\mathbb{V} \times \mathbb{I}_+}$.

$$\forall l \in [0, D] \cap \mathbb{Z} \quad \check{\boldsymbol{\Pi}}^{(l)} = \left(\begin{cases} \check{\pi}(\sigma^{(T)}, v, i_+) & d_{\mathcal{T}}(v) \leqslant l \\ 0 & d_{\mathcal{T}}(v) > l \end{cases}\right)_{(v,i_+) \in \mathbb{V} \times \mathbb{I}_+}$$

$$\check{\boldsymbol{\Pi}}^{(0)} = (\mathbf{1}_{v=v_0})_{(v,i_+) \in \mathbb{V} \times \mathbb{I}_+} \qquad \forall l \in [1, D] \cap \mathbb{Z} \quad \check{\boldsymbol{\Pi}}^{(l)} = \left(\left(\boldsymbol{L}^{(l)}\right)^\top \check{\boldsymbol{\Pi}}^{(l-1)}\right) \odot \check{\boldsymbol{S}} + \check{\boldsymbol{\Pi}}^{(l-1)} \tag{13}$$

For Equation 11 and Equation 13, we use in-place addition in our implementation to make sure only newly "visited" nodes are touched at each depth. This way, each node is only "visited" once during a single pass.

Let a vector $\check{\boldsymbol{\pi}} \in \mathbb{R}^{\mathbb{V}}$ be the terms in Equation 3 for counterfactual reach probabilities $\check{\pi} : \Sigma \times \mathbb{H} \to \mathbb{R}$.

$$\check{\boldsymbol{\pi}} = \left( \begin{cases} \begin{cases} \check{\pi}(\sigma^{(T)}, v, f_i(f_h(f_{Pa}(v)))) & f_h(f_{Pa}(v)) \in \mathbb{H}_+ \\ 0 & f_h(f_{Pa}(v)) \in \mathbb{H}_0 \end{cases} & v \in \mathbb{V}_+ \\ 0 & v = v_0 \end{cases} \right)_{v \in \mathbb{V}} = \left( \boldsymbol{M}^{(V, I_+)} \odot \check{\boldsymbol{\Pi}}^{(D)} \right) \boldsymbol{1}_{|\mathbb{I}_+|}$$

A vector $\widehat{\boldsymbol{\pi}} \in \mathbb{R}^{\mathbb{V}}$ representing the terms of the equation for "player" reach probabilities $\bar{\pi} : \Sigma \times \mathbb{H} \to \mathbb{R}$ in Equation 5 can be calculated identically but with $\widehat{\boldsymbol{S}}$ instead of $\check{\boldsymbol{S}}$ (defined in Equation 12) where

$$\widehat{\boldsymbol{S}} = \left( \begin{cases} 1 & \left( \boldsymbol{M}^{(V, I_+)} \right)_{v, i_+} = 0 \\ \boldsymbol{s}_v & \left( \boldsymbol{M}^{(V, I_+)} \right)_{v, i_+} = 1 \end{cases} \right)_{(v, i_+) \in \mathbb{V} \times \mathbb{I}_+}$$

$$\widehat{\boldsymbol{\pi}} = \left( \begin{cases} \begin{cases} \hat{\pi}(\sigma^{(T)}, v, f_i(f_h(f_{Pa}(v)))) & f_h(f_{Pa}(v)) \in \mathbb{H}_+ \\ 0 & f_h(f_{Pa}(v)) \in \mathbb{H}_0 \end{cases} & v \in \mathbb{V}_+ \\ 0 & v = v_0 \end{cases} \right)_{v \in \mathbb{V}}$$

### 3.2.2 AVERAGE STRATEGY PROFILE

The average strategy profile $\bar{\sigma}^{(T)} : \mathbb{Q} \to \mathbb{R}$ at an iteration $T$, formulated in Equation 10 and represented as a vector $\bar{\boldsymbol{\sigma}} \in \mathbb{R}^{\mathbb{Q}_+}$, can be updated from the previous iteration's $\bar{\sigma}^{(T-1)} : \mathbb{Q} \to \mathbb{R}$, represented as a vector $\bar{\boldsymbol{\sigma}}' \in \mathbb{R}^{\mathbb{Q}_+}$. For this, the "player" reach probabilities $\bar{\pi} : \Sigma \times \mathbb{H} \to \mathbb{R}$ (Equation 5), a restriction of which is represented by a vector $\bar{\boldsymbol{\pi}} \in \mathbb{R}^{\mathbb{H}_+}$, and their sums, a restriction of which is represented by a vector $\bar{\boldsymbol{\pi}}^{(\Sigma)} \in \mathbb{R}^{\mathbb{H}_+}$, must be calculated. The previous sums of counterfactual reach probabilities are denoted as a vector $\bar{\boldsymbol{\pi}}^{(\Sigma)\prime} \in \mathbb{R}^{\mathbb{H}_+}$.

$$\bar{\boldsymbol{\pi}} = \left( \bar{\pi}(\sigma^{(T)}, h_+) \right)_{h_+ \in \mathbb{H}_+} = \left( \boldsymbol{M}^{(H_+, Q_+)} \right) \left( \boldsymbol{M}^{(Q_+, V)} \right) \widehat{\boldsymbol{\pi}}$$

$$\bar{\boldsymbol{\pi}}^{(\Sigma)} = \left( \sum_{\tau=1}^{T} \bar{\pi}(\sigma^{(\tau)}, h_+) \right)_{h_+ \in \mathbb{H}_+} = \bar{\boldsymbol{\pi}}^{(\Sigma)\prime} + \bar{\boldsymbol{\pi}}$$

$$\bar{\boldsymbol{\sigma}} = \left( \bar{\sigma}^{(T)}(q_+) \right)_{q_+ \in \mathbb{Q}_+} = \bar{\boldsymbol{\sigma}}' + \left( \left( \boldsymbol{M}^{(H_+, Q_+)} \right)^{\top} \left( \bar{\boldsymbol{\pi}} \oslash \bar{\boldsymbol{\pi}}^{(\Sigma)} \right) \right) \odot \left( \boldsymbol{\sigma} - \bar{\boldsymbol{\sigma}}' \right) \tag{14}$$

### 3.2.3 NEXT STRATEGY PROFILE

Let a vector $\widetilde{\boldsymbol{r}} \in \mathbb{R}^{\mathbb{Q}_+}$ represent instantaneous counterfactual regrets $\tilde{r} : \Sigma \times \mathbb{Q}_+ \to \mathbb{R}$, defined in Equation 7, for a strategy profile $\sigma^{(T)}$ at an iteration $T$.

$$\widetilde{\boldsymbol{r}} = \left( \tilde{r}(\sigma^{(T)}, q_+) \right)_{q_+ \in \mathbb{Q}_+} = \left( \boldsymbol{M}^{(Q_+, V)} \right) \left( \check{\boldsymbol{\pi}} \odot \left( \left( \left( \boldsymbol{M}^{(V, I_+)} \right) \odot \left( \check{\boldsymbol{U}} - \boldsymbol{G}^{\top} \check{\boldsymbol{U}} \right) \right) \boldsymbol{1}_{|\mathbb{I}_+|} \right) \right)$$

Average counterfactual regrets $\bar{r}^{(T)} : \mathbb{Q}_+ \to \mathbb{R}$ in Equation 8 can be represented with a vector $\bar{\boldsymbol{r}} \in \mathbb{R}^{\mathbb{Q}_+}$. Let a vector $\bar{\boldsymbol{r}}' \in \mathbb{R}^{\mathbb{Q}_+}$ be the average counterfactual regrets at the previous iteration $\bar{r}^{(T-1)} : \mathbb{Q}_+ \to \mathbb{R}$.

$$\bar{\boldsymbol{r}} = \left( \bar{r}^{(T)}(q_+) \right)_{q_+ \in \mathbb{Q}_+} = \bar{\boldsymbol{r}}' + \frac{1}{T} \left( \widetilde{\boldsymbol{r}} - \bar{\boldsymbol{r}}' \right) \tag{15}$$

The clipped regrets are normalized to get a restriction of the next strategy profile $\sigma^{(T+1)} : \mathbb{Q}_+ \to \mathbb{R}$ from Equation 9 for (rational) player infoset-action pairs, represented as a vector $\boldsymbol{\sigma}'$.

$$\bar{\boldsymbol{r}}^{(+,\Sigma)} = \left( \sum_{a' \in A(h_+)} \left( \bar{r}^{(T)}(h_+, a') \right)^+ \right)_{(h_+,a) \in \mathbb{Q}_+} = \left( \boldsymbol{M}^{(H_+,Q_+)} \right)^\top \left( \left( \boldsymbol{M}^{(H_+,Q_+)} \right) \bar{\boldsymbol{r}}^+ \right)$$

$$\boldsymbol{\sigma}' = \left( \sigma^{(T+1)}(q_+) \right)_{q_+ \in \mathbb{Q}_+} = \left( \begin{cases} \left( \bar{\boldsymbol{r}}^+ \oslash \bar{\boldsymbol{r}}^{(+,\Sigma)} \right)_{q_+} & \left( \bar{\boldsymbol{r}}^{(+,\Sigma)} \right)_{q_+} > 0 \\ \left( \boldsymbol{\sigma}^{(T=1)} \right)_{q_+} & \left( \bar{\boldsymbol{r}}^{(+,\Sigma)} \right)_{q_+} = 0 \end{cases} \right)_{q_+ \in \mathbb{Q}_+}$$

## 4 BENCHMARKS

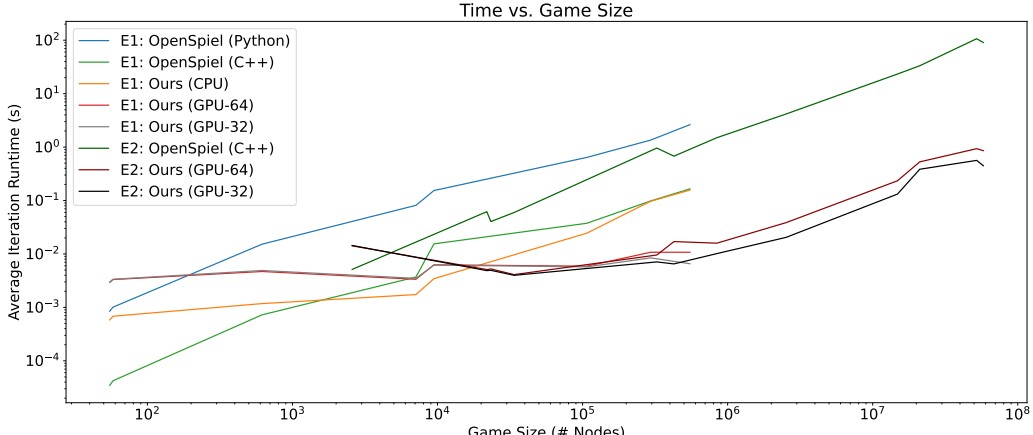

Figure 1: A log-log graph showing the average CFR iteration runtime with respect to the game size for both experiments. The four lines show the runtimes of the four benchmarked implementations. Note that the iteration time of each game does not depend solely on the number of nodes – the number of players and the number of infosets play a sizable role as well. In addition, for OpenSpiel implementations, the efficiency of how the game logic is implemented also matters as, on each iteration, their implementations traverse the game tree by generating new states online.

### 4.1 EXPERIMENT 1

We run 1,000 CFR iterations on 8 games of varying sizes implemented in Google DeepMind's OpenSpiel (Lanctot et al., 2020) (see Appendix C for more details) using their Python and C++ CFR implementations and our implementations (with a CPU or GPU backend). The games represent a diverse range of sizes from small (tiny Hanabi and Kuhn poker), medium (Kuhn poker (3-player), first sealed auction, and Leduc poker), to large (tiny bridge (2-player), liar's dice, and tic-tac-toe).

In our GPU implementation (written in Python), we use CuPy (Okuta et al., 2017) for GPU-accelerated matrix and vector operations with both 64-bit and 32-bit floating-point data types. Here, we only discuss the 32-bit implementation as it is generally faster than the 64-bit version. Note that we save both memory and runtime if single-precision floating point numbers are used (by roughly a factor of 2). We also simply run our implementation with NumPy (Harris et al., 2020) and SciPy (Virtanen et al., 2020) (i.e., without a GPU) which we refer to as our CPU implementation (with 64-bit floats). Our testbench computer contains an AMD Ryzen 9 3900X 12-core, 24-thread desktop processor, 128 GB memory, and Nvidia GeForce RTX 4090 24 GB VRAM graphics card.

The results vary depending on the size of the game being played, and are tabulated in Appendix A. The relationship between the game sizes and the runtimes of each implementation is shown more

clearly in the log-log graph in Figure 1. Note that our GPU implementation clearly scales better than both OpenSpiel's (Lanctot et al., 2020) and our CPU implementation.

#### 4.1.1 SMALL GAMES: TINY HANABI AND KUHN POKER

In small games like tiny Hanabi (55 nodes) and Kuhn poker (58 nodes), our CPU implementation shows modest gains over the OpenSpiel's (Lanctot et al., 2020) Python baseline (about 1.5 times faster for both). However, our GPU implementation is actually about 3.5 and 3.3 times slower for both compared to OpenSpiel's Python baseline. OpenSpiel's C++ baseline vastly outperforms all others by at least an order of magnitude. This suggests the overheads from GPU and Python make our implementation impractical for games of similarly small sizes.

#### 4.1.2 MEDIUM GAMES: KUHN POKER (3-PLAYER), FIRST SEALED AUCTION, AND LEDUC POKER

In medium-sized games like Kuhn poker (3-player) (617 nodes), first sealed auction (7,096 nodes), and Leduc poker (9,457 nodes), performance gains compared to OpenSpiel's (Lanctot et al., 2020) Python implementation can be observed for both our CPU (about 12.9, 46.8, and 44.6 times faster, respectively) and GPU implementation (about 3.1, 23.2, and 24.9 times faster, respectively). However, comparisons with OpenSpiel's C++ implementation are mixed. For Kuhn poker (3-player), OpenSpiel's C++ implementation is about 1.6 times faster than our CPU implementation and 6.8 times faster than our GPU implementation. But, for first sealed auction and Leduc poker, our CPU implementation is about 2.1 and 4.5 times faster, respectively, and our GPU implementation is about 1.1 and 2.5 times faster, respectively, than their C++ baseline. Here, while we begin to see our implementations outperform OpenSpiel's baselines, we see that our CPU implementation is faster than our GPU implementation. This suggests that, while the efficiency of our implementation overcomes the Python overhead, the remaining GPU overhead makes using a GPU less preferable than not.

#### 4.1.3 LARGE GAMES: TINY BRIDGE (2-PLAYER), LIAR'S DICE, AND TIC-TAC-TOE

In games like tiny bridge (2-player) (107,129 nodes), liar's dice (294,883 nodes), and tic-tac-toe (549,946 nodes), noticeable performance gains over OpenSpiel's (Lanctot et al., 2020) Python implementation can be observed for both our CPU (about 26.1, 13.9, and 16.8 times faster, respectively) and GPU implementation (about 111.8, 160.0, and 401.2 times faster, respectively). The same can be said for OpenSpiel's C++ implementation to a lesser degree: our CPU implementation is about 1.5, 1.0, and 1.1 times faster, respectively, and our GPU implementation is about 6.5, 11.6, and 25.2 times faster, respectively. Here, the performance benefits of utilizing a GPU are clear, and we predict that the differences will be even more pronounced for games of sizes larger than the ones explored.

#### 4.1.4 MEMORY USAGES

Table 1: The peak memory usage of the benchmark scripts of the 4 CFR implementations. For GPU implementations, peak usages of the process and the memory allocated by CuPy are shown.

| Implementation | | | | Peak Memory Usage (GB) |
|---|---|---|---|---|
| OpenSpiel | Python | | | 0.894 |
| | C++ | | | 0.145 |
| Ours | CPU | | | 2.863 |
| | GPU | 64-bit | Process | 3.169 |
| | | | CUDA | 0.273 |
| | | 32-bit | Process | 2.371 |
| | | | CUDA | 0.176 |

The peak memory usages of the benchmark scripts for Experiment 1 are shown in Table 1. Note that this is not exactly a fair comparison, as, in our implementations, we unnecessarily store the object representations of all states. By not doing so, further reduction in process memory usage would be possible. The allocated CUDA memory for each game is further analyzed in Appendix A.

## 4.2 EXPERIMENT 2

In order to see further scaling behavior or our GPU (both 64-bit and 32-bit versions) and OpenSpiel's C++ implementations, we run these implementations with much larger imperfect information games (12 battleship games of differing parameters) for 10 iterations each. The iteration times for Experiment 2 are tabulated in Appendix B. For games as large as up to over 57.9 million nodes, our GPU implementation performs up to 203.6 times faster than OpenSpiel's C++ baseline. Again, the relationship between the game sizes and the runtimes is shown in the log-log graph in Figure 1.

## 5 DISCUSSION

Our CFR implementations require a single complete game tree traversal during a setup phase to construct the game tree matrices. The time it takes to complete this one-time operation for the 20 games we test are tabulated in Appendix D. One advantage of our approach is that inefficient game logic implementation does not impact the runtime of our implementations as the game tree itself is encoded into matrices, whereas it can severely impact the performance of OpenSpiel's implementations which evaluate game logic during the traversal itself. This may explain the sizable gap in runtime between the speed of OpenSpiel's C++ implementation between the experiments even for the games of roughly the same sizes.

We only explore parallelizing the vanilla CFR algorithm, as proposed by Zinkevich et al. (2007). Later variants of CFR show improvements, namely in convergence speeds, which modify various aspects of the algorithm. The discounting techniques proposed by Brown and Sandholm can trivially be applied by altering Equation 14 and Equation 15. However, pruning techniques (Brown & Sandholm, 2015) would require non-trivial manipulations on the game-related matrices – possibly between iterations – problematic since updating CSR matrices is computationally expensive.

On each iteration, our implementation deals with the entire game tree and stores values for every node – impractical for extremely large games. In traditional implementations of CFR, while a complete recursive game tree traversal is carried out, counterfactual values are typically not stored for each node but instead for each infoset-actions. We demonstrate that it is possible to achieve a significant parallelization (and hence speedup) at a cost of higher memory usage. Intuitively, the root-to-leaf paths can be partitioned to construct subgraphs of which separate adjacency and submask matrices can be loaded and applied as necessary – a similar approach can be used for alternating player updates (Burch et al., 2019) and sampling variants (Lanctot et al., 2009).

Our approach provides an alternate way for CFR to be run on supercomputers. During the development of Cepheus (Tammelin et al., 2015), the game tree was chunked into a trunk and many subtrees, each of which was assigned to a compute node to be traversed independently. This introduced a bottleneck in the trunk as the subtree nodes (which depend on the trunk's results) must wait for the trunk calculation to complete during the downward pass, and wait again while the trunk uses the values returned by the subtrees during the upward pass. Our approach is simply a series of matrix/vector operations, and distributing this is a well-studied problem in systems.

In our GPU implementation, we used CuPy (Okuta et al., 2017) without any customizations in configurations and did not profile or probe into resource usage. A careful analysis of these for further optimizations will most likely yield further performance improvements.

## 6 CONCLUSION

We introduced our CFR implementation, designed to be parallelized by computing each iteration as dense and sparse matrix and vector operations and eliminating costly recursive tree traversal. While our goal was to run the algorithm on a GPU, the tight nature of our code also allows for a vastly more efficient computation even when a GPU is not leveraged. Our experiments on solving 20 games of differing sizes show that, in larger games, our implementation achieves orders of magnitude performance improvements over Google DeepMind's OpenSpiel (Lanctot et al., 2020) baselines in Python and C++, and predict that the performance benefit will be even more pronounced for games of sizes larger than those we tested. Addressing the memory inefficiency and incorporating the use of a GPU with non-vanilla CFR variants remains a promising avenue for future research.

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

## A  EXPERIMENT 1

In Experiment 1, we tested all four implementations – OpenSpiel's C++, OpenSpiel's Python, our CPU, and our GPU – in 8 commonly tested games (including a perfect-information game, tic-tac-toe, which CFR can be used to solve) by running CFR for 1,000 iterations.

Table 2: The average per-iteration runtimes (and the standard errors of the means, in brackets) of CFR implementations: reference OpenSpiel's and ours (with a CPU or a GPU). The performances of the fastest implementation for each game are bolded. The games are sorted by the number of nodes in the game tree and their names in the first column correspond exactly to the game name in Deepmind's OpenSpiel library.

| Game (in OpenSpiel) | Average CFR Iteration Runtime (milliseconds) | | | | |
| | OpenSpiel | | Ours | | |
| | Python | C++ | CPU | GPU | |
| | | | | 64-bit | 32-bit |
| --- | --- | --- | --- | --- | --- |
| `tiny_hanabi` | 0.851 (0.00) | **0.035 (0.00)** | 0.581 (0.00) | 2.958 (0.10) | 2.968 (0.10) |
| `kuhn_poker` | 1.011 (0.00) | **0.042 (0.00)** | 0.684 (0.00) | 3.319 (0.01) | 3.362 (0.00) |
| `kuhn_poker(players=3)` | 15.224 (0.01) | **0.725 (0.00)** | 1.177 (0.00) | 4.692 (0.01) | 4.906 (0.01) |
| `first_sealed_auction` | 81.226 (0.02) | 3.696 (0.01) | **1.736 (0.00)** | 3.355 (0.01) | 3.495 (0.00) |
| `leduc_poker` | 153.731 (0.19) | 15.444 (0.02) | **3.449 (0.00)** | 6.269 (0.01) | 6.178 (0.01) |
| `tiny_bridge_2p` | 640.783 (1.57) | 37.524 (0.25) | 24.513 (0.02) | 5.902 (0.01) | **5.732 (0.01)** |
| `liars_dice` | 1351.281 (8.39) | 98.109 (0.79) | 96.939 (0.07) | 10.766 (0.02) | **8.443 (0.01)** |
| `tic_tac_toe` | 2629.924 (11.04) | 165.389 (0.78) | 156.429 (0.15) | 10.756 (0.02) | **6.556 (0.00)** |

Table 3: The average per-iteration speedups or slowdowns in runtimes of our CFR implementations over reference OpenSpiel's. The positive values represent speedups and the negative values represent the slowdowns. The games are sorted by the number of nodes in the game tree and their names in the first column correspond exactly to the game name in Deepmind's OpenSpiel library. A similar table showing the original raw runtime values is Table 2.

| Game (in OpenSpiel) | Average Speedup or Slowdown (times) | | | | | |
| | OpenSpiel's Python | | | OpenSpiel's C++ | | |
| | Our CPU | Our GPU | | Our CPU | Our GPU | |
| | | 64-bit | 32-bit | | 64-bit | 32-bit |
| --- | --- | --- | --- | --- | --- | --- |
| `tiny_hanabi` | 1.5 | -3.5 | -3.5 | -16.8 | -85.2 | -85.5 |
| `kuhn_poker` | 1.5 | -3.3 | -3.3 | -16.1 | -78.3 | -79.3 |
| `kuhn_poker(players=3)` | 12.9 | 3.2 | 3.1 | -1.6 | -6.5 | -6.8 |
| `first_sealed_auction` | 46.8 | 24.2 | 23.2 | 2.1 | 1.1 | 1.1 |
| `leduc_poker` | 44.6 | 24.5 | 24.9 | 4.5 | 2.5 | 2.5 |
| `tiny_bridge_2p` | 26.1 | 108.6 | 111.8 | 1.5 | 6.4 | 6.5 |
| `liars_dice` | 13.9 | 125.5 | 160.0 | 1.0 | 9.1 | 11.6 |
| `tic_tac_toe` | 16.8 | 244.5 | 401.2 | 1.1 | 15.4 | 25.2 |

The raw values and speedups (or slowdowns) are tabulated in Table 2 and Table 3, respectively.

The total allocated CUDA memory by CuPy (Okuta et al., 2017) in our GPU implementation to solve each game through CFR in Experiment 1 is plotted in Figure 2 and tabulated in Table 4. Note

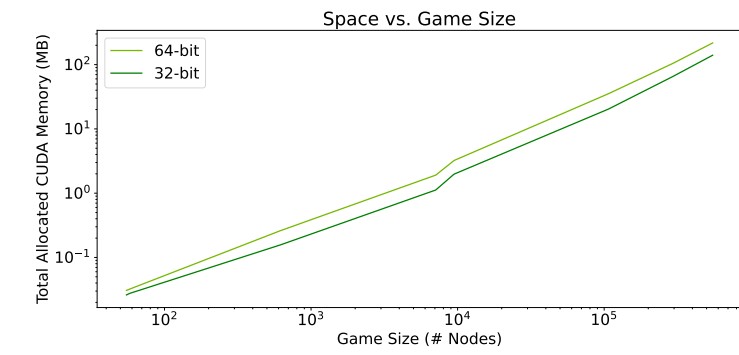

Figure 2: A log-log graph showing the total allocated CUDA memory by our GPU implementation for each game tested in Experiment 1.

Table 4: The total allocated CUDA memory during CFR iterations for each game tested in Experiment 1. The games are sorted by the number of nodes in the game tree and their names in the first column correspond exactly to the game name in Deepmind's OpenSpiel library.

| Game (in OpenSpiel) | Total Allocated CUDA Memory (MB) | |
|---|---|---|
| | Double-Precision (64-bit) | Single-Precision (32-bit) |
| tiny_hanabi | 0.031 | 0.026 |
| kuhn_poker | 0.032 | 0.028 |
| kuhn_poker(players=3) | 0.261 | 0.157 |
| first_sealed_auction | 1.911 | 1.121 |
| leduc_poker | 3.224 | 1.978 |
| tiny_bridge_2p | 35.327 | 20.373 |
| liars_dice | 104.731 | 65.520 |
| tic_tac_toe | 217.263 | 139.977 |

that noticeable improvement in memory usage is seen when 32-bit floating-point numbers are used instead of 64-bit floating-point numbers.

While exploitability is a concept only valid for 2-player zero-sum games (and our method is also applicable to and tested on non-2-player general-sum games), we ran our algorithm again (separately from the benchmarks) with exploitabilities for Experiment 1 (on GPU 64-bit implementation). These are plotted in Figure 3. Note that the convergence behavior of CFR is already well-known, and our contribution is not about optimizing the convergence metrics like exploitability as most past CFR works have been about. We calculated exploitabilities purely for the sanity testing of our implementation.

## B    EXPERIMENT 2

In Experiment 2, we tested two implementations – OpenSpiel's C++ and our GPU – in 12 very large battleship games (up to over 57.9 million nodes) by running CFR for 10 iterations. As these games are large, we loaded the game without storing all state objects into memory (instead, we use Python integers).

The raw values and speedups (or slowdowns) are tabulated in Table 5 and Table 6, respectively. Note that our 32-bit GPU implementation performs about 2.8 times slower for the zeroth battleship game, but is up to 203.6 times faster (for the seventh battleship game).

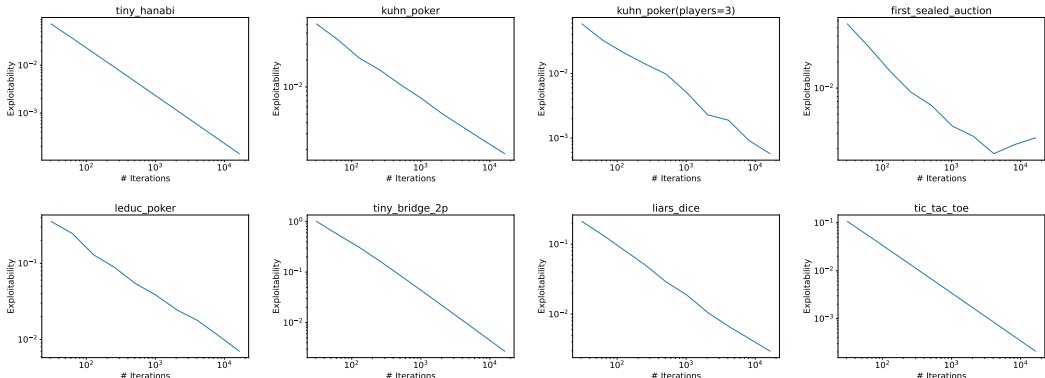

Figure 3: Log-log graphs of exploitabilities for each game tested using our GPU implementation for the first 16,384 iterations. Note that some of these games are not 2-player zero-sum games where the concept of exploitability is not well-defined. These are only analyzed for games we tested in Experiment 1.

Table 5: The average per-iteration runtimes (and the standard errors of the means, in brackets) of CFR implementations: reference OpenSpiel's (C++) and ours (with a GPU). The performances of the fastest implementation for each game are bolded. The games are sorted by the number of nodes in the game tree. The comma-separated parameters represent the board width, board height, ship sizes, and number of shots, respectively. The ship values were set to a list of ones.

| Game (Named by Us) | Parameters | Average CFR Iteration Runtime (milliseconds) | | |
|---|---|---|---|---|
| | | OpenSpiel (C++) | Ours (GPU) | |
| | | | 64-bit | 32-bit |
| Battleship-0 | 2,2,[1],2 | **5.134 (0.26)** | 14.197 (10.21) | 14.393 (10.57) |
| Battleship-1 | 2,2,[1;2],2 | 61.807 (4.50) | 5.099 (0.20) | **4.880 (0.14)** |
| Battleship-2 | 2,2,[1],3 | 40.392 (4.21) | 5.247 (0.21) | **4.939 (0.13)** |
| Battleship-3 | 2,3,[1],2 | 59.662 (3.53) | 4.104 (0.27) | **3.983 (0.13)** |
| Battleship-4 | 2,2,[1;2],3 | 959.262 (122.18) | 9.558 (0.42) | **7.094 (0.32)** |
| Battleship-5 | 3,3,[1],2 | 676.297 (36.66) | 17.034 (9.80) | **6.510 (0.20)** |
| Battleship-6 | 2,3,[1],3 | 1499.620 (120.59) | 15.913 (0.83) | **10.156 (0.43)** |
| Battleship-7 | 3,4,[1],2 | 4161.539 (159.38) | 38.554 (2.36) | **20.437 (1.04)** |
| Battleship-8 | 4,4,[1],2 | 23262.634 (1034.90) | 233.995 (9.59) | **131.795 (3.48)** |
| Battleship-9 | 2,3,[1],4 | 33245.108 (4039.16) | 528.117 (25.22) | **384.525 (5.81)** |
| Battleship-10 | 3,3,[1;2],2 | 106613.962 (6977.96) | 933.899 (48.55) | **563.536 (9.90)** |
| Battleship-11 | 4,5,[1],2 | 90346.083 (5783.18) | 856.541 (59.31) | **446.369 (10.94)** |

## C   GAME PROPERTIES

Table 7 gives details (e.g. number of nodes, terminal nodes, infosets, actions, and players) about the games we solve during both our experiments, and Table 8 shows the sparsities of the mask matrices when the discrete games we explore are converted into our desired format.

## D   SETUP

In order to use our implementation, the game tree must first be transformed into sparse matrices encoding the game rules. This requires a single complete game tree traversal. Note that this is a one-time operation performed prior to running our CFR implementations. Table 9 shows the time it takes to serialize each discrete game from OpenSpiel (Lanctot et al., 2020).

Table 6: The average per-iteration speedups or slowdowns in runtimes of our GPU-CFR implementation over reference OpenSpiel's (C++). The positive values represent speedups and the negative values represent the slowdowns. The games are sorted by the number of nodes in the game tree. A similar table showing the original raw runtime values is Table 5.

| Game | Average Speedup or Slowdown (times) | |
|---|---|---|
| | Double-Precision (64-bit) | Single-Precision (32-bit) |
| Battleship-0 | -2.8 | -2.8 |
| Battleship-1 | 12.1 | 12.7 |
| Battleship-2 | 7.7 | 8.2 |
| Battleship-3 | 14.5 | 15.0 |
| Battleship-4 | 100.4 | 135.2 |
| Battleship-5 | 39.7 | 103.9 |
| Battleship-6 | 94.2 | 147.7 |
| Battleship-7 | 107.9 | 203.6 |
| Battleship-8 | 99.4 | 176.5 |
| Battleship-9 | 63.0 | 86.5 |
| Battleship-10 | 114.2 | 189.2 |
| Battleship-11 | 105.5 | 202.4 |

Table 7: The 8 (Experiment 1) plus 12 (Experiment 2) games tested in our benchmark and relevant statistics: number of nodes, terminal nodes, infosets, actions, and (rational) players. The games were grouped by the experiment they belonged to, and then sorted by the number of nodes in the game tree.

| Game | # Nodes | # Terminals | # Infosets | # Actions | # Players |
|---|---|---|---|---|---|
| tiny_hanabi | 55 | 36 | 8 | 3 | 2 |
| kuhn_poker | 58 | 30 | 12 | 3 | 2 |
| kuhn_poker(players=3) | 617 | 312 | 48 | 4 | 3 |
| first_sealed_auction | 7,096 | 3,410 | 20 | 11 | 2 |
| leduc_poker | 9,457 | 5,520 | 936 | 6 | 2 |
| tiny_bridge_2p | 107,129 | 53,340 | 3,584 | 28 | 2 |
| liars_dice | 294,883 | 147,420 | 24,576 | 13 | 2 |
| tic_tac_toe | 549,946 | 255,168 | 294,778 | 9 | 2 |
| Battleship-0 | 2,581 | 1,936 | 210 | 8 | 2 |
| Battleship-1 | 21,877 | 16,384 | 1,970 | 10 | 2 |
| Battleship-2 | 23,317 | 17,488 | 2,514 | 8 | 2 |
| Battleship-3 | 33,739 | 28,116 | 1,118 | 12 | 2 |
| Battleship-4 | 324,981 | 243,712 | 46,962 | 10 | 2 |
| Battleship-5 | 426,556 | 379,161 | 5,915 | 18 | 2 |
| Battleship-6 | 843,739 | 703,116 | 33,518 | 12 | 2 |
| Battleship-7 | 2,529,949 | 2,319,120 | 19,154 | 24 | 2 |
| Battleship-8 | 14,811,409 | 13,885,696 | 61,698 | 32 | 2 |
| Battleship-9 | 21,093,739 | 17,578,116 | 1,005,518 | 12 | 2 |
| Battleship-10 | 52,081,183 | 46,294,416 | 204,980 | 24 | 2 |
| Battleship-11 | 57,920,421 | 55,024,400 | 152,402 | 40 | 2 |

Table 8: The sparsities of sparse matrix constants in our implementation. CUDA's (Nickolls et al., 2008) cuSPARSE "library targets matrices with sparsity ratios in the range between 70%-99.9%" (cuS). Our values fall under this recommended range. We project that the matrices for games not tested in our work will typically have similar sparsity values as those we test.

| Game | Sparsities (%) | | | |
|---|---|---|---|---|
| | $M^{(Q_+,V)}$ | $M^{(H_+,Q_+)}$ | $L^{(l)}$ (Average) | $G$ |
| tiny_hanabi | 96.4 | 87.5 | 99.6 | 98.2 |
| kuhn_poker | 96.6 | 91.7 | 99.7 | 98.3 |
| kuhn_poker(players=3) | 99.0 | 97.9 | 99.9+ | 99.8 |
| first_sealed_auction | 99.5 | 95.0 | 99.9+ | 99.9+ |
| leduc_poker | 99.9+ | 99.9 | 99.9+ | 99.9+ |
| tiny_bridge_2p | 99.9+ | 99.9+ | 99.9+ | 99.9+ |
| liars_dice | 99.9+ | 99.9+ | 99.9+ | 99.9+ |
| tic_tac_toe | 99.9+ | 99.9+ | 99.9+ | 99.9+ |
| Battleship-0 | 99.9+ | 99.9+ | 99.9+ | 99.9+ |
| Battleship-1 | 99.9+ | 99.9+ | 99.9+ | 99.9+ |
| Battleship-2 | 99.9+ | 99.9+ | 99.9+ | 99.9+ |
| Battleship-3 | 99.9+ | 99.9+ | 99.9+ | 99.9+ |
| Battleship-4 | 99.9+ | 99.9+ | 99.9+ | 99.9+ |
| Battleship-5 | 99.9+ | 99.9+ | 99.9+ | 99.9+ |
| Battleship-6 | 99.9+ | 99.9+ | 99.9+ | 99.9+ |
| Battleship-7 | 99.9+ | 99.9+ | 99.9+ | 99.9+ |
| Battleship-8 | 99.9+ | 99.9+ | 99.9+ | 99.9+ |
| Battleship-9 | 99.9+ | 99.9+ | 99.9+ | 99.9+ |
| Battleship-10 | 99.9+ | 99.9+ | 99.9+ | 99.9+ |
| Battleship-11 | 99.9+ | 99.9+ | 99.9+ | 99.9+ |

Table 9: The times it took to convert OpenSpiel's discrete games into sparse matrices in our implementations. These games include those tested in any one of our experiments.

| Game | Setup Time (seconds) |
|---|---|
| tiny_hanabi | 0.427 |
| kuhn_poker | 0.010 |
| kuhn_poker(players=3) | 0.070 |
| first_sealed_auction | 0.734 |
| leduc_poker | 1.051 |
| tiny_bridge_2p | 11.795 |
| liars_dice | 34.264 |
| tic_tac_toe | 62.521 |
| Battleship-0 | 0.706 |
| Battleship-1 | 2.939 |
| Battleship-2 | 2.995 |
| Battleship-3 | 6.660 |
| Battleship-4 | 48.742 |
| Battleship-5 | 53.252 |
| Battleship-6 | 117.131 |
| Battleship-7 | 254.096 |
| Battleship-8 | 6736.911 |
| Battleship-9 | 2280.204 |
| Battleship-10 | 5446.435 |
| Battleship-11 | 5470.082 |

