# OpenReview forum: "GPU-Accelerated Counterfactual Regret Minimization"
_ICLR.cc/2025/Conference — Submitted to ICLR 2025_

### Official Review · Reviewer_hUL5 · 2024-10-26

**Soundness:** 2
**Presentation:** 2
**Contribution:** 2
**Rating:** 5
**Confidence:** 4

**Summary:**

This manuscript discussed about how to leverage GPU’s computation power on sparse matrix multiplication to improve the computation speed for large scale games.

**Strengths:**

* Leveraging GPUs to improve the computation for game solving is still under-explored and this manuscript has a solid step towards it.

**Weaknesses:**

* This is a more engineering-driven paper without lots of scientific contribution.
* I’m not quite sure about where the benefit is from. See the questions.

**Questions:**

* Is OpenSpiel’s implementation paralleled? I think this can have a lot of differences in the performance. Only comparing with the benchmark implementation can be limited.
* You showed there is a trade-off for memory efficiency and computation time. However I don’t see there are any reasons on this if we store every sparse matrix within the corresponding format. Can you further explain this?

---

> ### Author Response · Authors · 2024-11-18
> **Thank you for your review!**
>
> >This is a more engineering-driven paper without lots of scientific contribution.
>
> It is true that, in our paper, we do not propose anything theoretically novel. Rather, we introduce methods to accelerate the highly relevant CFR algorithm to take advantage of GPUs for orders of magnitude speedup.
>
> We would like to point out that one of the relevant subject areas mentioned by ICLR "infrastructure, software libraries, hardware, etc." fits our work quite well, and ICLR has published systems papers before.
>
> >Is OpenSpiel’s implementation paralleled? I think this can have a lot of differences in the performance. Only comparing with the benchmark implementation can be limited.
>
> No, OpenSpiel's implementation is not parallelized, but our CPU implementation isn't either. Yet, we show that our CPU implementation is still faster than that of OpenSpiel.
>
> Our formulation (matrix/vector calculations) reduces the number of control-flow instructions. We also see that introducing parallelization with GPUs vastly improves the performance compared to the baselines.
>
> We chose OpenSpiel as it offers a diverse variety of games, and is widely used in the field of CFR to benchmark different CFR variants.
>
> >You showed there is a trade-off for memory efficiency and computation time.
>
> What we wanted to emphasize is that we store a corresponding value for each node as opposed to each infoset-action pair (like OpenSpiel does). There are usually many more nodes than infoset-actions.
>
> We discuss potential ways our approach can be augmented to overcome this limitation (e.g., only loading a partition of the game tree into memory).
>
> We hope this addresses all your reservations and questions about the paper.

---

> > ### Comment · Reviewer_hUL5 · 2024-11-21
> > **Thanks for the clarification.**
> >
> > I personally think that eliminating the control-flow instructions with matrix computation should be an interesting view for CFR-based algorithms but this generally suffers from a tradeoff between parallelism and memory. The proposal is interesting but for me it is not a significant technical breakthrough. I would leave the decision to other reviewers and area chairs.

---

> > > ### Author Response · Authors · 2024-11-22
> > >
> > > Thank you for your feedback! We appreciate your consideration.

---

### Official Review · Reviewer_Vhx6 · 2024-10-28

**Soundness:** 3
**Presentation:** 3
**Contribution:** 2
**Rating:** 5
**Confidence:** 4

**Summary:**

This paper proposes a GPU implementation of the CFR algorithm. It stores the game tree structure in matrices and performs a series of matrix operations to compute reach probabilities, expected payoffs, and other relevant information level by level, thereby avoiding the costly tree traversal process. By leveraging GPU acceleration for these matrix operations, the approach significantly enhances CFR’s speed, albeit at the expense of increased memory usage.

**Strengths:**

* The paper is well-written, and the notations are clearly defined.
* The GPU implementation significantly boosts the speed of the CFR algorithm, and the difference from the OpenSpiel implementation grows as the game size increases.
* The code is open-sourced, making it easy to apply to games in OpenSpiel.

**Weaknesses:**

This is not the first approach to implementing CFR on GPUs. For example, the thesis by Reis [1] and the report by Weng [2] also present GPU implementations of CFR. Additionally, DeepStack [3] [4] constructs the look-ahead tree and runs CFR on GPUs. These works and this paper use a similar approach, computing relevant information in CFR level by level and accelerating it with GPUs. However, the paper does not compare or discuss these related works.

[1] Reis J. A gpu implementation of counterfactual regret minimization. Master Thesis, University of Porto, 2015.

[2] Weng L. Counterfactual regret minimization on gpu.
https://cent.felk.cvut.cz/courses/GPU/archives/2020-2021/W/rudolja1/, 2021

[3] Moravčík M, Schmid M, Burch N, et al. Deepstack: Expert-level artificial intelligence in heads-
up no-limit poker. Science, 2017, 356(6337): 508-513.

[4] Schmid M. DeepStack for Leduc Hold'em. https://github.com/lifrordi/DeepStack-Leduc, 2017

**Questions:**

* What is the space complexity of the method?
* Can this method be used to accelerate the calculation of exploitability?

Minor Suggestions:

* It will help to improve clearity by adding a symbol table and an example figure.
* It will help to enhance readability by keeping the symbol descriptions consistent with CFR literatures, such as using $I$ for an information set and $h$ for a history node.

---

> ### Author Response · Authors · 2024-11-14
> **Thank You for Your Review!**
>
> There are **significant** differences in how GPUs are used in our work versus the works that you brought up.
>
> * We frame the problem of CFR as a series of sparse matrix/vector operations, and the utilization of GPUs for this task is very well studied.
> * Reis and Rudolf (not Weng) wrote the CFR implementation **including iterative/recursive tree traversal on CUDA**. (edit: added iterative).
> * DeepStack's GPU usage is for their **neural network evaluation**. They still perform costly recursive traversal for the lookahead operation (not parallelized). The "Student of Games" and "ReBeL" algorithms also use GPU for their NN, and these particular algorithms are mentioned in our introduction.
>
> They or their connection to GPUs were not mentioned due to space constraints and their (perceived) irrelevance to our approach. With that said, we will mention them in our revision if the reviewer feels strongly about it.
>
> >Space complexity
>
> It's O(VI) where V is the number of nodes and I is the number of players. (Edit: changed from linear w.r.t. game size)
>
> >accelerate the calculation of exploitability?
>
> Yes, our approach can be used. But, we did not explore it in this work.
>
> We hope this addresses some of your concerns.

---

> ### Comment · Reviewer_Vhx6 · 2024-11-14
>
> Thank you for the quick response.
>
> Using Rudolf’s implementation as an example, the iterative CFR described in section 4.3 does not require recursive tree traversals during CFR iterations. From my perspective, the implementation of iterative CFR is similar to the method proposed in the paper.
>
> Both methods use a recursive tree traversal initially to compute the relationship between adjacent levels in the tree, setting up the structure for subsequent CFR iterations. During CFR iterations, both methods compute utilities by levels in descending order and compute the reach probabilities levels in ascending order, and no recursive tree traversals are needed. The dense matrices defined in the paper, such as $\Pi, U,$ and $\bar{\sigma}$, correspond to the data structures stored in Rudolf’s code. Additionally, the sparse matrix/vector operations defined in the paper serve as the formal definitions for the for-loops in Rudolf’s code. The part of the matrix multiplication that utilizes GPU acceleration in Rudolf’s code is almost the section of the for-loops that is parallelized.
>
> In summary, I don't think there is much difference between the two methods

---

> > ### Author Response · Authors · 2024-11-14
> > **Thank you for your response!**
> >
> > >In summary, I don't think there is much difference between the two methods
> >
> > We still maintain that our approaches to tree traversal are substantially different.
> >
> > ### Rudolf (Line 92 to 136 and 51 to 90): https://github.com/janrvdolf/gpucfr/blob/master/src/gpucfr.cu
> >
> > * For each node in the game tree, Rudolf's code, starting from that node, **moves up the game tree** toward the root.
> > * In other words, **for each node, Rudolf makes O(H) extra visits** where H is the maximum height of the tree.
> > * Since the height of the tree depends on the game, the number of nodes Rudolf visits each iteration is **linearithmic (n log n) in practice and O(n^2) in the worst case.**
> > * He does this because each node must take account of all nodes in the subtree rooted at that node.
> >
> > ### Ours
> >
> > * We parallelly calculate each node level by level.
> > * Each node is only visited **once**.
> > * We managed to parallelize tree traversal while maintaining linear time complexity.
> >
> > In both of our approaches, through tree traversal, relevant data is aggregated to each information set. From this point, the calculation is vastly easier to parallelize (which you noticed was similar).
> >
> > >The part of the matrix multiplication that utilizes GPU acceleration in Rudolf’s code is almost the section of the for-loops that is parallelized.
> >
> > * Yes, the equation we use in some parts is identical because we implement the **same** algorithm!
> > * If these are different, then one of us is not implementing CFR correctly.
> >
> > In addition, we would like to stress that Rudolf's code only supports 2-player games (See Line 125). Our code supports any finite number of players. This makes the implementation much more complex as we must keep track of counterfactual reach probabilities, counterfactual utilities, utilities, etc. for each player.
> >
> > I hope this resolves any misunderstanding on our and their approaches.

---

> > > ### Comment · Reviewer_Vhx6 · 2024-11-15
> > >
> > > Apologied for the typo. I mixed up the authores. I was referring to Reis's thesis, not Rudolf's report.
> > >
> > > In Figures 17 and 18 of Reis's thesis, the pseudocode of the iterative version of CFR is described. It computes the reach probabilities level-by-level in ascending order and compute utilities level-by-level in descending order. It does not require recursive tree traversal. In fact, it appers to perform operations similar to those in Equations 26 and 21 in your paper.
> > >
> > > > The part of the matrix multiplication that utilizes GPU acceleration in Reis’s code is almost the section of the for-loops that is parallelized.
> > >
> > > Since the computation of one level depends on other levels, the main GPU acceleration in CFR comes from parallelizing the computations within each level. The sparse matrix/vector operations achieve this implicitly by accelrating matrix multiplication using the GPU, while the pseudocode achieves this explictly by assigning multiple CPU/GPU cores to distribute the for-loops ("#parga omp parallel for" in Line 2 of Figures 17 and 12).

---

> ### Author Response · Authors · 2024-11-15
> **Thank you very much for your response!**
>
> We have to admit that we were not familiar with Reis's work, as it was not published in a peer-reviewed venue and their code is unavailable. (Edit: note that "[authors] may be excused for not knowing about papers not published in peer-reviewed conference proceedings or journals" as per the 2025 ICLR AC guide.)
>
> It is now clear that Reis has already done the level-by-level updating for CFR. Thank you for bringing this to our attention.
>
> With that said, we firmly believe that this does not disqualify our work.
>
> * The idea of using level graphs for graph calculations is a very widespread technique. We simply thought this was not done for CFR before.
> * The code shown in the thesis would not compile. Also, his code cannot be found online.
>   * Figure 12 Line 4's missing semicolon.
>   * Figure 18 Line 8's mismatched square braces.
> * We reduce everything to matrix/vector operations which is much more efficient on GPUs due to opcodes optimized for those purposes. On the other hand, their implementation requires each thread to carry out more generalized kernel instructions for updating at each node.
> * Their compatibility (and exploration) of games is limited to custom variants of poker. In our analysis, we explore selections of discrete games that are available on OpenSpiel (widely used for research in CFR).
>
> With that said, we do see a need to update our manuscript to mention the works you brought up. We will work to revise the manuscript and update it by the end of this week.
>
> Again, thank you for the catch, and we hope this addresses all your reservations about the paper.

---

> > ### Comment · Reviewer_Vhx6 · 2024-11-16
> >
> > > With that said, we firmly believe that this does not disqualify our work.
> >
> > Given Reis’s report, the paper’s contributions are liimted. This is not the first paper to use GPUs to accelerate CFR, to compute nodes in the same level in parallel, and to avoid recursive tree traversal. The main contribution is the proposal of a series of matrix/vector operations, which, from my perspective,  serves as a formalism of the for-loops in Reis's code.

---

> ### Author Response · Authors · 2024-11-16
>
> We deeply appreciate your response!
>
> We never claimed to be the first one to have discovered a level-by-level updating technique in our paper -- using level graphs is a very widely known technique in graph theory. Plus, in our implementation, level-by-level updates (graph traversal) are only used to calculate counterfactual reach probabilities and expected utilities. It is not used for counterfactual reach probabilities, counterfactual regrets, and next/average strategy profiles.
>
> Even considering level-by-level updates are made by Reis for the graph traversal part, our discovery that a tight implementation with the ability to utilize matrix/vector opcodes exists for CFR remains novel (thus eliminating a ``large number of control flow statements'', a limitation mentioned by Reis) (including parts unrelated to graph traversal).
>
> Aside from Reis's screenshots containing syntax errors and omitting details such as calculating counterfactual regrets, strategy profiles, and counterfactual reach probabilities, his code does not handle chance nodes, decision nodes, and terminal nodes separately. We strongly doubt that his work can be reproduced to work in practice.
>
> We also stress that our implementation, unlike Reis's, is much more interesting in that it is compatible with OpenSpiel, widely used by the community while being implemented in pure Python. Plus, we release our code to the public.
>
> Again, we really appreciate your deep engagement with the paper. While we hope that we addressed some of your concerns, we understand if your mind is already made up about the significance of our work. (Edit: we have addressed them in the new revision)

---

> > ### Comment · Reviewer_Vhx6 · 2024-11-26
> >
> > I appreciate the authors for the detailed response. I’m raising my score accordingly, but I feel the contributions are insufficient for a higher score.

---

> > > ### Author Response · Authors · 2024-11-26
> > >
> > > Thank you for your feedback! We appreciate the reviewer for their consideration.

---

### Official Review · Reviewer_eX7u · 2024-11-03

**Soundness:** 2
**Presentation:** 2
**Contribution:** 2
**Rating:** 3
**Confidence:** 1

**Summary:**

This paper studies hardware acceleration for counterfactual regret minimization.

**Strengths:**

The acceleration seems quite significant as the author claimed.

**Weaknesses:**

Well, it is unclear to me if this paper fits well for ICLR since there is no new algorithm / methodology / theory proposed. It may fit more to ML system venue. The benchmark selected (Game in OpenSpiel) is less known. I will suggest show improvements on more common benchmarks. I have to admit I do not have sufficient GPU hardware background to evaluate this paper.

**Questions:**

see above

---

> ### Author Response · Authors · 2024-11-15
> **Thank you for your review!**
>
> >Well, it is unclear to me if this paper fits well for ICLR since there is no new algorithm / methodology / theory proposed.
>
> It is true that, in our paper, we do not propose anything theoretically novel. Rather, we introduce methods to accelerate the highly relevant CFR algorithm to take advantage of GPUs for orders of magnitude speedup.
>
> >It may fit more to ML system venue.
>
> We would like to point out that one of the relevant subject areas mentioned by ICLR "infrastructure, software libraries, hardware, etc." fits our work quite well, and ICLR has published systems papers before.
>
> >The benchmark selected (Game in OpenSpiel) is less known
>
> OpenSpiel's games are widely used in the field of CFR to benchmark different CFR variants. Our contribution of the accompanying software (in the supplementary file) is a software library that is not only compatible with discrete games on OpenSpiel but also shows significant acceleration.
>
> Again, thank you for your review, and we hope this addresses some of your reservations about our paper.

---

### Official Review · Reviewer_dyt9 · 2024-11-04

**Soundness:** 3
**Presentation:** 3
**Contribution:** 3
**Rating:** 3
**Confidence:** 4

**Summary:**

This paper aims to frame the CFR algorithm as a sequence of matrix operations (Think GraphBlas).

**Strengths:**

This paper is interesting because it tries to solve two problems at the same time:
- APIs like [GraphBLAS](https://graphblas.org/) have successfully represented graph algorithms as a sequence of BLAS-like operations over semirings. This paper tries to do the same for CFR.
- It's not obvious how GPUs, the powerhouse of deep learning, can be used to accelerate game solving (other than calling neural networks). This paper tries to solve this gap.

**Weaknesses:**

Overall, this paper tries to aim for a best-of-both-worlds approach: low coding effort and high performance.
Instead, it ends up with an exposition that is somehow less clear than the original CFR paper, benchmarks that don't inspire confidence, and the resulting algorithm seems to be not very flexible and requires major efforts to do the simplest changes like going from simultaneous to alternating variants of CFR.

- The open spiel codebase is not an example of a performant CFR implementation, it is meant to be extremely generic. Comparing with more reasonable implementations is necessary (for instance the Cepheus codebase for Leduc, there are other codebases as well).
- The games tested are extremely tiny. The largest game (tic tac toe) takes less than 3 seconds to traverse on a pure Python implementation and is a perfect information game. It's not clear what a perfect information game is doing there. The number of nodes in Appendix A is the number of infosets + terminals (i.e., there is only one node per infoset), and looking at the code, `tic_tac_toe.py` is not a dark variant of tic-tac-toe.
- The updates are batched and masked. Effectively, for each depth $d$, a (sparse) matrix multiply is done but only the output at depth $d$ is stored. As a matter of efficiency, this method becomes more and more wasteful as the depth of the game increases.
- Modifying the algorithm seems extremely expensive and cumbersome. (see  line 515)
- The computations presented here are probably not very GPU friendly and a lot of FLOPS are probably wasted (Yet this has not been analyzed).
- The paper does not include any exploitability results.

**Questions:**

- What was the inspiration for using CuPy?
- Why are there no exploitability results?
- How long does it take to setup the game tree?
- Were 32 or 64-bit floats used? Were tensor cores enabled?

- 022- The abstract is used in the introduction
- 037- GPU is defined twice
- 062- What is the philosophical reason for making the nature player have infosets? I find this very nonstandard
- 163- While CFR can find approximate coarse correlated equilibrium, the formulation presented in this paper does not. The reason is that it is not the product of average strategies that converges to the set of CCEs, but rather the average product.
- 175- The introduction should include the CFR theorem ($r^{(T)}(i_{+,j}\in\mathbf{I}_{+})$ is bounded by the sum of the local regrets)
- 201- Case in point in that this paper tries to inflate its notation, $l\in[1, D]\cap \mathbf{Z}$ is just weird.
- 292- Hadamard product was not defined. I would argue that it is not a common notation and should be defined explicitly.
- 397- It is not clear what purpose Figure 1 serves in the paper.
- 432- Why is there no error bar on the time measurements?
- 515- I understand that pruning the tree may be expensive (surely a very big downside) but why is alternating CFR hard to implement?
- 523- "Our approach is also much easier (and likely more efficient) to be run on supercomputers compared to previous methods" This statement is not backed by anything in this paper, see Weaknesses.
- 651- the number of infosets for liars_dice seems to be wrong, it should be 24576 not 24583 (or 12288 if we remove infosets with only one action).

---

> ### Author Response · Authors · 2024-11-14
> **Thank You for Your Very Detailed Review!**
>
> We would like to address some points that were brought up.
>
> >The open spiel codebase is not an example of a performant CFR implementation, it is meant to be extremely generic.
>
> * Our implementation is also designed to be extremely generic, compatible with all discrete games on OpenSpiel plus more. As far as we are aware, no other software solution provides a CFR implementation with a wide array of games.
> * OpenSpiel provides multiple CFR implementations, one of which -- their C++ implementation -- is supposed to be their performant version.
> * OpenSpiel is widely used by researchers. A faster library compatible with OpenSpiel can be useful.
>
> >The games tested are extremely tiny. [...] It's not clear what a perfect information game is doing there. [...] `tic_tac_toe.py` dark variant of tic-tac-toe.
>
> * These games (except tic-tac-toe) are widely used to test the convergence speed of algorithms in the field of CFR [1].
> * CFR can absolutely be used to solve perfect information games (like tic-tac-toe).
> * In our tests, we only used `gpugt/games/open_spiel.py`.
>
> >this method becomes more and more wasteful as the depth of the game increases.
>
> We don't quite understand this. Our experiments indicate that our method scales well. Also, note that the growth of the depth is logarithmic w.r.t. the game size.
>
> >Modifying the algorithm seems extremely expensive and cumbersome
>
> >not very flexible and requires major efforts to do the simplest changes like going from simultaneous to alternating variants of CFR.
>
> Some modifications are trivial, as stressed in the paper. With that said, we acknowledge the non-trivial nature of some modifications (e.g., pruning) in the paper is a limitation. Please note that the implementation difficulty of going from vanilla CFR to an alternating player variant (e.g., CFR+) is quite cumbersome even when parallelization is not considered.
>
> >not very GPU friendly and a lot of FLOPS are probably wasted
>
> Again, empirical experiments show our method is still quite performant. We acknowledge the lack of further profiling in the paper.
>
> >The paper does not include any exploitability results.
>
> The convergence behavior of CFR is already very well known and we do not propose any new algorithm or variant. In addition, exploitability is strictly a 2-player concept, and our method generalizes to (and is tested on) games whose number of players exceeds 2. With that said, it can be added to the Appendix if the reviewer feels strongly about it.
>
> >Why [...] CuPy?
>
> Similar API as NumPy which allows us to swap between them to see performance with and without parallelization.
>
> >How long does it take to setup the game tree?
>
> * Our code performs a single game tree traversal, followed by filling in the matrices.
> * It takes from 0.015 seconds (Kuhn poker) to 67.951 seconds (tic-tac-toe).
> * The times can be added to the Appendix if the reviewer feels strongly about it.
>
> >Were 32 or 64-bit floats used? Were tensor cores enabled?
>
> * 64-bit floats were used.
> * Tensor cores were not used.
>
> >reason for making the nature player have infosets
>
> * We found it to be more consistent.
> * It helped us derive the matrix/vector equations.
>
> >it is not the product of average strategies that converges to the set of CCEs, but rather the average product.
>
> Our Equation 8 is identical to Equation 4 in Zinkevich et al [2].
>
> >The introduction should include the CFR theorem
>
> Some CFR papers omit this. But, we can add it in our revision.
>
> >why is alternating CFR hard to implement?
>
> At the time of writing the paper, it seemed to require updating sparse matrix values mid-way. But, now, we understand that this can be implemented by creating separate masks for each actor during setup.
>
> >This statement is not backed by anything in this paper
>
> * We did not elaborate due to space constraints.
> * Tammelin et al. [3] parallelized CFR by chunking the game tree with a trunk and many subtrees. Each subtree and trunk was assigned to a node that traversed the assigned tree.
> * Our approach is simply a series of sparse matrix/vector operations, and distributing this is a well-studied problem in systems.
>
> >the number of infosets for liars_dice seems to be wrong
>
> * We considered nature's decision node to form a singleton infoset, hence the 16 infosets for Kuhn poker (instead of 12), 1093 for Leduc poker (instead of 936), et cetera.
>
> We apologize for the long response... there was a lot to cover. We hope we addressed some of your concerns. Thank you for your stylistic suggestions.
>
> [1] H. Xu et al., 2024. Dynamic Discounted Counterfactual Regret Minimization. URL: https://opnreview.net/forum?id=6PbvbLyqT6
>
> [2] M. Zinkevich et al., 2007. Regret Minimization in Games with Incomplete Information. URL: https://poker.cs.ualberta.ca/publications/NIPS07-cfr.pdf
>
> [3] O. Tammelin et al., 2015. Solving Heads-up Limit Texas Hold’em. URL: https://poker.cs.ualberta.ca/publications/2015-ijcai-cfrplus.pdf

---

> ### Comment · Reviewer_dyt9 · 2024-11-18
>
> > Their C++ implementation is supposed to be their performant version.
>
> Performant is a relative to the python implementation. As far as I can tell, no large game was solved with this CFR implementation.
>
> > OpenSpiel is also widely used by researchers... a faster library compatible with OpenSpiel's games can be useful.
>
> OpenSpiel is widely used to sample actions or extract game trees, I do not believe the CFR implementation is widely used.
>
> > These games are widely used to test the convergence speed of algorithms in the field of CFR [1].
>
> And I still claim that these games are tiny and not meaningful, specially for performance considerations. If traversing the game tree takes less than 1 second per iterations, is it even worth optimizing? This is not a denial of your method, rather that your experiment are simply not enough to convince the utility of your method.
>
> > CFR can absolutely be used to solve perfect information games.
>
> That's not the point, It's that the speedup obtained in TTT is not really representative of any real usecase of CFR. Alpha-Beta pruning can find an optimal solution in a single pass of the game tree.
>
> > We don't quite understand this. Our experiments indicate that our method scales well. Also, note that the growth of the depth is logarithmic w.r.t. the game size.
>
> The logarithmic growth is under the assumption that game tree grows uniformly, which is not necessarily true. At every depth, you updated a vector that is #infosets sized not #infosets at depth sized. IMHO, separating the buffers by depth makes sense, while you are not visiting every node $H$ times like the other implementation method mentioned here, you are visiting every node max_depth times.
>
> > The convergence behavior of CFR is already very well known and we do not propose any new algorithm or variant. With that said, it can be added to the Appendix if the reviewer feels strongly about it
> > We argue that the reviewer is conflating the general expectations in CFR papers with what is relevant to our specific contribution.
>
> The exploitability is not a sanity check for CFR, but for your codebase.
>
> > > it is not the product of average strategies that converges to the set of CCEs, but rather the average product.
>
> > Our Equation 8 is identical to Equation 4 in Zinkevich et al [2].
>
> That's not the issue, the issue is that the way the regret buffers are updated. The code does not obtain a CCE. I would ask that you either change the phrase, or make a note that it would require keeping the average product of policies (instead of the product of the average that you do).
>
> Thank you for the following clarifications!
>
> > Our code performs a single game tree traversal, followed by filling in the matrices.
>
> > It takes from 0.015 seconds (Kuhn poker) to 67.951 seconds (tic-tac-toe).
>
> > The times can be added to the Appendix if the reviewer feels strongly about it.
>
> That would be great!
>
> > 64-bit floats were used. Tensor cores were not used.
>
> I think it would be a good addition (albeit to your discretion) to add runtime (and exploitability) results for 32-bit variants as well as 4090s are ~64 times faster using 32bit floats tho It is not clear if that is actually the bottleneck given the need for lots of control using sparse matrices.
>
> > In our tests, we only used gpugt/games/open_spiel.py.
>
> > We considered nature's decision node to form a singleton infoset, hence the 16 infosets for Kuhn poker (instead of 12), 1093 for Leduc poker (instead of 936), et cetera.
>
> > We found it to be more consistent.
>
> > It helped us derive the matrix/vector equations.

---

> ### Author Response · Authors · 2024-11-20
> **Thank you for your response & New Results (Larger Games, Exploitabilities, etc.)**
>
> >OpenSpiel is widely used to sample actions or extract game trees, I do not believe the CFR implementation is widely used.
>
> OpenSpiel was lead-developed by Dr. Marc Lanctot, who was part of Dr. Michael Bowling's group in UAlberta. A lot of the AI/RL papers from Google DeepMind (where Dr. Lanctot now works) use OpenSpiel. There is evidence that Dr. Tuomas Sandholm's group from CMU also uses OpenSpiel [1, 2]. Ordinarily, we would not like to appeal to authority like this, but we think it's important context that some of the most influential and respected research groups in CFR use OpenSpiel.
>
> We would like to point out that, in general, there is a lack of good benchmarks for CFR performance.
>
> >That's not the point, It's that the speedup obtained in TTT is not really representative of any real usecase of CFR. Alpha-Beta pruning can find an optimal solution in a single pass of the game tree.
>
> While we feel our initial set of games tested is sufficient, the new revision contains new experimental results from imperfect information games of up to over 57.9 million nodes on which our GPU implementation achieves speedups of up to 114.2 times.
>
> >If traversing the game tree takes less than 1 second per iterations, is it even worth optimizing?
>
> CFR typically runs for thousands or tens of thousands of iterations. Our method saves substantial time for even the "tiny" games that you mentioned.
>
> >At every depth, you updated a vector that is #infosets sized not #infosets at depth sized.
>
> In our code, we do in-place addition "+=" which only touches newly visited nodes (See Lines 265 and 292 of `gpugt/algorithms/counterfactual_regret_minimization.py`). This way, we only "visit" each node once in a single pass. With that said, a naive implementation of the formula we show will not do that though (thank you for catching this). We made that clear in the new revision.
>
> >The exploitability is not a sanity check for CFR, but for your codebase.
> >That's not the issue, the issue is that the way the regret buffers are updated.
>
> Our new revision implements non-parallelized exploitability calculation (results in Appendix A). During the process, we found an issue in our average strategy formula. It's been resolved and our code was found to behave identically with OpenSpiel's base CFR implementation (without the default alternating player updates) for select games.
>
> In summary, the new revision addresses most of your stylistic suggestions, your criticisms, and answers to your questions.
>
> Considering the effort put into the rebuttals, we kindly request the reviewer to consider our latest updates.
>
> [1] Perolat J, et al. From poincaré recurrence to convergence in imperfect information games: Finding equilibrium via regularization. International Conference on Machine Learning, 2021. URL: https://proceedings.mlr.press/v139/perolat21a/perolat21a-supp.pdf
>
> [2] McAleer S., et al. ESCHER: Eschewing Importance Sampling in Games by Computing a History Value Function to Estimate Regret. The Eleventh International Conference on Learning Representations, 2023. URL: https://web.mit.edu/~gfarina/www/2023/escher_iclr23/2206.04122.pdf

---

> > ### Author Response · Authors · 2024-11-26
> >
> > Given that the rebuttal deadline is fast approaching, we would like to remind the reviewer of our latest changes and request their feedback.

---

> > > ### Comment · Reviewer_dyt9 · 2024-11-27
> > >
> > > > Given that the rebuttal deadline is fast approaching, we would like to remind the reviewer of our latest changes and request their feedback.
> > >
> > > I think ICLR sends enough notifications as is 🙃.
> > >
> > > re: ESCHER
> > >
> > > I think my point stands, ESCHER does not use the CFR implementation of OpenSpiel.
> > >
> > > > In our code, we do in-place addition "+=" which only touches newly visited nodes (See Lines 265 and 292 of gpugt/algorithms/counterfactual_regret_minimization.py). This way, we only "visit" each node once in a single pass. With that said, a naive implementation of the formula we show will not do that though (thank you for catching this). We made that clear in the new revision.
> > >
> > > That is still not visiting every node once, the dynamic indexing is not free. I maintain that having one buffer per level makes more sense. That being said, it's somewhat hard to claim that it is expensive w/o any benchmark.
> > >
> > > > CFR typically runs for thousands or tens of thousands of iterations. Our method saves substantial time for even the "tiny" games that you mentioned.
> > >
> > > There are two counter arguments: you are comparing one CPU core to one GPU, I think a CPU core is around 100 times more expensive than a GPU. It's easy to argue that you could launch 100 CFRs in parallel.
> > > The other argument is that most large games get less than 2000 iterations: Cepheus did ~1700, Supremus does 1000 iterations.
> > >
> > > Citing Supremus is important. It does ~1000 iterations (on the gpu) in less than a second. Granted it is not directly comparable with the method proposed here, still it serves at an ideal to aim for.
> > >
> > > > Our Equation 8 is identical to Equation 4 in Zinkevich et al [2].
> > >
> > > And Zinkevich does not mention CCEs. The statement is correct it just doesn't match the rest of the paper or the equation that appears right after it.
> > >
> > > re: the new experiments
> > >
> > > I think they are a great addition, still the bar for the baseline (OpenSpiel's CFR) is pretty low. I tried running the code with liars dice 2 dices and 4 faces but other than warning messages like `OpenSpiel exception: /project/open_spiel/games/liars_dice/liars_dice.cc:279 player >= 0`, the code would take forever just to build the tree (it took 12 minutes and 70GB of memory, iterating over the game itself only takes 1.5 minutes). Running 2 dices 3 faces takes ~0.03 seconds per iteration with 50s spent creating the CFR matrix and game tree, In contrast the rebel codebase can do 1000 iterations in less than 3 seconds.
> > >
> > > re: style
> > >
> > > The tables do not follow the style guideline for ICLR. The table caption must come *before* the table.
> > >
> > >
> > >
> > > Cepheus: http://poker.srv.ualberta.ca/
> > >
> > > Supremus: Zarick, Ryan, et al. "Unlocking the potential of deep counterfactual value networks." arXiv preprint arXiv:2007.10442 (2020).
> > >
> > > rebel: https://github.com/facebookresearch/rebel

---

> ### Author Response · Authors · 2024-11-27
>
> Thank you for your swift response!
>
> >I think my point stands, ESCHER does not use the CFR implementation of OpenSpiel.
>
> The point we are making is that researchers who were heavily involved with SOTA CFR research (including those who created pioneering AI agents for large-scale games) both developed and used OpenSpiel.
>
> >still the bar for the baseline (OpenSpiel's CFR) is pretty low.
>
> As we said, there is a lack of good benchmarks for CFR that are compatible with general games.
>
> >That is still not visiting every node once, the dynamic indexing is not free.
>
> Conceptually we "visit" each node once, as the underlying matrix only has to only update or incorporate non-zero locations, and hence its time complexity (for +=) depends on the number of non-zero elements being added. Plus, the Hadamard product depends on the number of non-zero elements in the operand with fewer non-zero elements (as only the intersection of non-zero elements needs to be considered). Also, the matrix multiplication with a level graph only considers nodes relevant to that level. Making these runtime behavior assumptions is acceptable in systems literature.
>
> >There are two counter arguments: you are comparing one CPU core to one GPU.
>
> Yes, but we also run benchmarks with our CPU implementation (not parallelized) which is still faster than the baseline for games beyond certain sizes.
>
> >It's easy to argue that you could launch 100 CFRs in parallel
>
> One can't simply "launch" 100 CFRs in parallel. We talk about how previous works parallelized it in discussions (and in this comment chain) and the difficulty in doing so. Also, your argument is analogous to saying that there is no point in using GPUs for deep learning, as people can just use 100 CPUs.
>
> >Citing Supremus is important. It does ~1000 iterations (on the gpu) in less than a second. [...] In contrast the rebel codebase can do 1000 iterations in less than 3 seconds.
>
> These are depth-limited search variants. They only do a light lookahead instead of a full game tree traversal. They are simply not comparable. Plus, given these are completely different algorithms one cannot do a one-to-one comparison with the number of iterations. Note that the number of iterations depends on the target exploitability measure.
>
> >Cepheus did ~1700 [iterations]
>
> Note that it took them 68 days to calculate the approximate solution [1]. Our method has the potential for substantial speedups (still, doing this is obviously a substantial engineering challenge and out of the scope of this paper).
>
> >And Zinkevich does not mention CCEs. The statement is correct it just doesn't match the rest of the paper or the equation that appears right after it.
>
> Again, Zinkevich explores 2-player games. In this case, the average policy is an NE. But, you seem to acknowledge this is correct.
>
> Do you mind clarifying how the rest of the paper does not match Equation 10? We actually compared our average policy output with that of OpenSpiel's implementation (without alternating player updates) for a selection of games and found them to be "identical" save some floating point inaccuracies.
>
> >re: the new experiments [...] I think they are a great addition
>
> Thank you.
>
> >code would take forever just to build the tree
>
> Thank you for taking an interest in running our code!
>
> This setup stage of defining the game as matrices (which we did not bother optimizing) should be considered strictly separate from the CFR algorithm (which we did optimize) -- if done once, they can be shared and reused anytime by others.
>
> With that said, even if we do consider this one-time operation into the runtime, the overall time we can save during iteration is substantial in the case of large games and more than makes up for the game tree setup.
>
> >The tables do not follow the style guideline for ICLR. The table caption must come before the table.
>
> Thank you for pointing this out. We will incorporate this in our manuscript.
>
> [1] O. Tammelin et al. Solving Heads-Up Limit Texas Hold’em, 2015. URL: https://www.ijcai.org/Proceedings/15/Papers/097.pdf

---

> > ### Comment · Reviewer_dyt9 · 2024-11-27
> >
> > > As we said, there is a lack of good benchmarks for CFR that are compatible with general games.
> >
> > Problem specific solvers do provide a goal to aim for, including them will make this a better paper (in my opinion).
> >
> > > Do you mind clarifying how the rest of the paper does not match Equation 10? We actually compared our average policy output with that of OpenSpiel's implementation (without alternating player updates) for a selection of games and found them to be "identical" save some floating point inaccuracies.
> >
> > I think there is a misunderstanding, the statement is correct but the machinery (either theoretic or code) does not support finding CCEs. Yes, there exists a variant of this paper/codebase that does find CCEs, but this is not that variant.
> >
> > > Conceptually we "visit" each node once, as the underlying matrix only has to only update or incorporate non-zero locations, and hence its time complexity (for +=) depends on the number of non-zero elements being added. This is an acceptable assumption to be made in systems literature.
> >
> > Unfortunately, I'm neither a systems person, nor this is a systems conference, nor this paper was submitted  to systems related area of research.
> >
> > Due to the architectural particularities of GPUs the inplace update kernel has to read the mask array. Thread divergences will make this task much more expensive that it needs to be. Benchmarking is really necessary.
> >
> > > One can't simply "launch" 100 CFRs in parallel. We talk about how previous works parallelized it in discussions (and in this comment chain).
> >
> > But that's often what happens, the practitioner either has many benchmarks, or has to solve many subproblems.
> >
> > > These are depth-limited search variants
> >
> > Supremus can solve rivers that are bigger than most examples in this paper in a fraction of a second.
> >
> > > Note that it took them 68 days to calculate the approximate solution [1]. Our method has the potential for substantial speedups (still, doing this is obviously a substantial engineering challenge and out of the scope of this paper).
> >
> > You are missing the point. Nobody seems to be doing tens of thousands of iterations as far as I can tell.
> >
> > I think that this statement can only be made after comparing the performance with Cepheus on a medium sized poker game. Cepheus can run in single node mode (with parallelism) and is extremely flexible with the variants of poker it supports.
> >
> > > This setup stage of defining the game as matrices (which we did not bother optimizing) should be considered strictly separate from the CFR algorithm (which we did optimize) -- if done once, they can be shared and reused anytime by others.
> >
> > I will add my two cents that storing large trees as maps and sets is probably not a good idea. Serializing the 70GB of memory required to store the game tree for 2d4f liars dice is probably going to take longer than recreating it from scratch on the average HPC cluster I've used.

---

> ### Author Response · Authors · 2024-11-27
>
> Again, we greatly appreciate your timely response and your vigorous review. Our manuscript has improved substantially.
>
> I suggest we do not discuss CFR benchmarks anymore. The AC just weighed in on this debate.
>
> >Yes, there exists a variant of this paper/codebase that does find CCEs, but this is not that variant.
>
> "CFR minimizes external regret (Zinkevich et al. 2007), so it converges to a coarse correlated equilibrium." [1]
>
> >Due to the architectural particularities of GPUs the inplace update kernel has to read the mask array. Thread divergences will make this task much more expensive that it needs to be. Benchmarking is really necessary.
>
> Yes, you are right, but, it is standard practice to disregard specific hardware optimization/behaviors when discussing the time/space complexity of algorithms. The entire field of CS is built on top of this assumption.
>
> >You are missing the point. Nobody seems to be doing tens of thousands of iterations as far as I can tell.
>
> Brown and Sandholm [1] (up to 32K) and Lockhart et al (up to 100K) do. But, it is true that they primarily study the convergence behaviors in their paper (which our software can be used for). Generally, the number of iterations for real-life use cases are chosen to reach exploitability irrelevant to the human timescale in real-life gameplay. And yes, these happened to be in the range of thousands of iterations for most large games solved as of recent. Still, our method has already shown improvement even in this scale.
>
> >Serializing the 70GB of memory required to store the game tree for 2d4f liars dice is probably going to take longer than recreating it from scratch on the average HPC cluster I've used.
>
> I disagree, the actual matrices are up to of size ~7.822 GB... the current code stores each game-object representation in memory which is obviously unnecessary. (Edit: the new more efficient game definition function avoids this.) Again, our focus was not on the optimization of the game tree setup.
>
> With that aside, do you mind acknowledging in the AC's comment that you were able to run 2d4f liar's dice? The AC is under the impression that it failed. The warning is because we ask for information set on a chance node. Since we ignore nature's "information sets" in our implementation, it does not affect our later calculation.
>
> Again, thank you very much.
>
> [1] N. Brown and T. Sandholm. Solving Imperfect-Information Games via Discounted Regret Minimization, 2019. URL: https://arxiv.org/pdf/1809.04040
>
> [2] Lockhart et al. Computing Approximate Equilibria in Sequential Adversarial Games by Exploitability Descent, 2019. URL: https://arxiv.org/pdf/1903.05614

---

> > ### Author Response · Authors · 2024-11-29
> >
> > We would like to remind the reviewer of our latest changes (please see Summary of Changes (2)), and request their feedback. We hope our last comment and the latest changes to the manuscript bring some closure to a number of items that were brought up.

---

### Official Review · Reviewer_zSia · 2024-11-04

**Soundness:** 3
**Presentation:** 3
**Contribution:** 2
**Rating:** 5
**Confidence:** 3

**Summary:**

The paper presents a GPU-accelerated approach for Counterfactual Regret Minimization (CFR), a class of no-regret learning algorithms widely used in solving large-scale imperfect information games like poker. Traditionally, CFR algorithms rely on recursive tree traversal, which limits their efficiency. This work proposes a restructured CFR algorithm that converts the operations into a series of dense and sparse matrix and vector computations, making it highly parallelizable on GPUs at the cost of increased memory usage.

The study concludes that the proposed approach could serve as a foundational step for highly scalable CFR on supercomputing infrastructures, enabling faster solutions to complex game-theoretic problems.

**Strengths:**

Originality: The paper introduces a creative approach by reformulating Counterfactual Regret Minimization (CFR) as matrix operations suitable for GPU processing. This novel restructuring allows a highly parallelizable version of CFR, which has not been extensively explored in existing CFR literature.

Efficiency in Design: By avoiding recursive tree traversal, the implementation achieves substantial speed gains, especially in larger games, demonstrating an efficient design choice that effectively leverages GPU hardware.

Thorough Empirical Evaluation: The paper evaluates the new approach on a diverse set of games with varying complexity and size, rigorously benchmarking it against established OpenSpiel baselines in both Python and C++. This experimental breadth strengthens the validity of its claims about speedup and scalability.

Significant Potential for Scalability: The approach is well-suited for large-scale games, with experiments showing up to 352.5 times faster performance than OpenSpiel’s Python baseline and 22.2 times faster than its C++ counterpart, especially promising for future work on supercomputing platforms or even more extensive imperfect information games.

**Weaknesses:**

Originality Limitations: Although innovative, the paper applies GPU parallelization to the vanilla CFR algorithm, which is somewhat limited in novelty given the existence of other CFR variants that incorporate modern enhancements (e.g., CFR+ or discounting techniques). A broader implementation encompassing these would increase the relevance of this work.

Limited Exploration of Advanced CFR Variants: The paper does not explore compatibility with modern CFR variants, such as sampling-based or discounting techniques, which are widely used in state-of-the-art game-solving algorithms. This omission reduces the approach’s applicability in advanced game AI contexts.

High Memory Requirements: The matrix-based reformulation, while speeding up calculations, results in high memory consumption, especially for larger games. This trade-off is not thoroughly analyzed or discussed, particularly regarding potential bottlenecks for memory-constrained systems, which limits practical usability.

Narrow Application Scope: The results indicate that this GPU implementation is unsuitable for small games, where it can actually perform worse than CPU implementations due to overhead. This could reduce its perceived impact and limits its practicality in domains that deal with a wide range of game sizes.

**Questions:**

N/A

---

> ### Author Response · Authors · 2024-11-18
> **Thank you for your review!**
>
> >Originality Limitations: [...] vanilla CFR algorithm [...] is [...] limited in novelty given the existence of other CFR variants that incorporate modern enhancements
>
> >Limited Exploration of Advanced CFR Variants: [...] The paper does not explore compatibility with modern CFR variants, such as sampling-based or discounting techniques
>
> Our work of parallelizing CFR represents a first step that opens the door to being augmented to support alternative CFR variants (e.g., CFR+). In our discussion section, We do bring up potential ways these variants (e.g., CFR+, discounting techniques) can be parallelized in our fashion (some of which are quite trivial). Technically, sampling variants can be thought of as only applying a small partition of the game tree, which we discuss briefly. We delegate further exploration as potential future work that builds on top of our work.
>
> >High Memory Requirements: [...] results in high memory consumption, especially for larger games. This trade-off is not thoroughly analyzed or discussed
>
> The linear space complexity with respect the the size of the game tree is something we acknowledge as a limitation. Note that we also discuss potential ways to reduce memory consumption such as only loading a partition of the game tree into memory, as necessary. Also, in our paper, we analyze the memory consumption during the game-solving of each discrete game.
>
> >The results indicate that this GPU implementation is unsuitable for small games, where it can actually perform worse than CPU implementations due to overhead.
>
> This is a standard limitation of GPU acceleration in any use case.

---

### Author Response · Authors · 2024-11-18
**Common Responses and Changes in our Revision**

We would like to thank all the reviewers for their reviews and take this opportunity to give generalized comments and summarize the revisions we made to our manuscript.

We are happy that most reviewers agree that the degree of GPU acceleration in our work is "substantial"/"significant"/"solid" (Reviewers zSia, eX7u, Vhx6, and hUL5). Even Reviewer dyt9 who took an issue with our benchmark (more on this later) at least found our paper "interesting". New experimental results (at the request of Reviewer dyt9) for games orders of magnitudes larger than the ones previously tested show an even greater speedup of up to 114.2 times faster than OpenSpiel's baselines.

Reviewers hUL5 and eX7u expressed sentiment that the "engineering-driven"/"system"s nature of our contribution does not fit well for ICLR. We would like to point out that ICLR has published numerous systems papers before and "infrastructure, software libraries, hardware, etc." is one of the relevant subject areas explicitly mentioned by ICLR.

Reviewer Vhx6 brought up that GPUs have been leveraged before for CFR [1-2] or CFR-inspired algorithms like DeepStack [3]. In our new revision and the discussion chain, we argue how our transformation of CFR into linear algebra operations (for GPU usage) greatly differs from the works the reviewer cited. Note that works by Reis and Rudolf the reviewer emphasizes are not peer-reviewed and/or self-published.

Reviewers dyt9 and eX7u questioned if the use of OpenSpiel's games and baselines for benchmarks was sufficient. We argue that it is, as selections of its games (that we also test on) have been used by a wide variety of past CFR/imperfect-information game AI papers [4-5], and our solution is designed to be as generic as OpenSpiel's. In general, there is a lack of good benchmarks for CFR performance. Our new revision contains experimental results from games of sizes orders of magnitudes larger (at the request of Reviewer dyt9 who felt the games were not large enough).

Reviewer dyt9 pointed out the lack of exploitability results. We argued that the reviewer is conflating the general expectations in CFR papers with what is relevant to our specific contribution. With that said, we calculated exploitabilities (as sanity tests) and added the log-log plots in our latest revision (Appendix A). During the process, we found a bug in our code and formulation when it came to average strategy calculation which we fixed in the new revision (and supplementary).

We also incorporated several stylistic suggestions and answers to reviewers' questions. Again, we would like to thank the reviewers for pointing them out. Overall, we believe our paper has been significantly strengthened following this review process.

With that aside, we provided detailed responses to all of the reviewers' concerns and questions. We hope to have addressed them adequately.

[1] Reis J. A gpu implementation of counterfactual regret minimization. Master Thesis, University of Porto, 2015. URL: https://repositorio-aberto.up.pt/handle/10216/83517

[2] Weng L. Counterfactual regret minimization on gpu, 2021. URL: https://cent.felk.cvut.cz/courses/GPU/archives/2020-2021/W/rudolja1/

[3] Moravčík M, et al. Deepstack: Expert-level artificial intelligence in heads-up no-limit poker. Science, 2017, 356(6337): 508-513.

[4] Perolat J, et al. From poincaré recurrence to convergence in imperfect information games: Finding equilibrium via regularization. International Conference on Machine Learning, 2021.

[5] McAleer S., et al. ESCHER: Eschewing Importance Sampling in Games by Computing a History Value Function to Estimate Regret. The Eleventh International Conference on Learning Representations, 2023. URL: https://web.mit.edu/~gfarina/www/2023/escher_iclr23/2206.04122.pdf

---

> ### Author Response · Authors · 2024-11-27
> **Summary of Changes (1)**
>
> * New experiment with imperfect-information games of sizes up to over 57.9 million (orders of magnitudes larger than previous) -- we call this Experiment 2. (Requested by Reviewer dyt9)
> * Exploration of the unpublished prior works. (Requested by Reviewer Vhx6)
> * Expanded discussions, including about running on supercomputers (requested by Reviewer dyt9) and advantages of transformation into linear algebra operations.
> * Reformulation of player reach probabilities (part of average strategy profile) and updated results.
> * Reorganization and cleanup of the Appendix into the following sections: Exp. 1, Exp. 2 (new), game properties, and setup times (new).
>   * Exp. 1 contains new exploitability plots.
>   * Game properties count infosets in a more typical way.
>   * Raw iteration times (with stderr, requested by Reviewer dyt9), table of sparsities, and CUDA memory usage plot were moved to the Appendix due to space constraints.
> * The 16-page-long expansions of equations in the Appendix were removed as we felt it was distracting and the fonts were too small. The rebuttal revision uploaded on 24 Nov 2024 contains them. They can be restored to camera-ready if requested.
> * Stylistic changes
>   * Improved formula font sizing
>   * Replaced "information set" with "infoset"
>   * Improved writing in various places

---

> ### Author Response · Authors · 2024-11-27
> **Summary of Changes (2)**
>
> Since our last summary, we made a number of changes to our submission.
>
> * Additional test with 32-bit floating-point format (previously only 64-bit was done). This leads to faster runtime and lower memory usage (by roughly a factor of 2). Our GPU algorithm now performs 203.6 times faster than OpenSpiel's performant C++ algorithm. (Requested by Reviewer dyt9 and AC).
> * Added more memory and time-efficient game definition (during setup) in our supplementary code.

---

### Comment · Area_Chair_i3Jj · 2024-11-27

Dear all,

First of all, thanks to everyone for engaging in a polite discussion about the merits of the paper.

I have noticed there have been many messages exchanged, so I thought it would be productive to try to summarize the most contested points and focus the discussion further for the next few days before the (extended) author discussion closes.

First, though, let me get a few things out of the way:
- to Reviewers hUL5 and eX7u: I agree with the authors that engineering work does fit under the umbrella of welcomed contributions at ICLR
- I agree with the authors that (in my experience) "there is a lack of good benchmarks for CFR performance."
- I agree with the authors that "authors may be excused for not knowing about papers not published in peer-reviewed conference proceedings or journals" as per reviewing guidelines.

Overall, it seems that one of the main points of disagreement between Reviewer dyt9 and the authors is about whether comparison against OpenSpiel is meaningful. As the reviewers pointed out, OpenSpiel is one de-facto standard for entry-level game-solving primitives and a well-established library. On the other hand, I think that Reviewer dyt9 has a point that comparing a sophisticated GPU implementation against OpenSpiel's "vanilla" Python implementation is apples-to-oranges and not necessarily informative. I agree with Reviewer dyt9 that the implementations of CFR that ended up used in breakthrough results were much more tuned than the Python implementation of OpenSpiel. That being said, most of those implementations are not publicly available (or not general purpose), and the authors' contribution could help people today. So, it seems that the major point of disagreement is not about whether the author's contribution is welcome (it is!), but rather whether the evaluation of said contribution is thorough and convincing. In this regard, the following points have been brought up:
- The games used in the benchmarks might be too small. The authors seem to have added at least one game to address this.
- Exploitability plots were missing. This was a good suggestion by Reviewer dyt9, which helped the authors catch a minor bug.
- OpenSpiel might be spending the majority of time constructing internal bookkeeping that's not directly correlated with running CFR iterations.
- The authors' implementation is not yet capable of implementing alternating CFR.
- Certain low-hanging benchmarks have been left out (such as turning on tensor cores, or ablating the choice of floating point format).
- Reviewers Vhx6 and hU5 seemed to believe the technical novelty is insufficient. However, some of the references they brought up seem to refer to unpublished work with no publicly available code, so I think it is unfair to hold it against the authors.

As things stand, from my point of view, the main merit of the authors' contribution is that it would provide the community with an open-source implementation of CFR that is significantly faster than the default Python one in OpenSpiel, while retaining compatibility with OpenSpiel games. That being said, I think that the reviewers have brought up several important points. For example, the inability to solve Liar's Dice with two dice and four faces each seems a bit of a letdown, given that the game is not that large. In addition, I agree that certain ablations might make evaluating this ML systems paper more thorough.

If I may add on top of the reviewers, one point I did not fully understand about the implementation (but I might have completely missed) is whether CFR is implemented as an algorithm on the game tree, or on each player's sequence-form polytope. The latter can be significantly smaller, and the benefit is important in games such as poker that admit fast gradient and best-response computation (in time proportional to the number of information sets as opposed to nodes in the game tree; e.g., Johanson et al. 2011). Implementations based on the sequence form are, in my experience, significantly faster and easier to parallelize.

That being said, I want to thank again both the authors and the reviewers for their vigorous engagement in this paper. While not all parties might be in agreement on all the points, it is clear that significant effort has been put into the lengthy back-and-forth discussion.

---

> ### Author Response · Authors · 2024-11-27
>
> Dear AC,
>
> Thank you for feedback and your balanced position regarding our debate! Overall, we agree with most of your points. The below contain some clarifications.
>
> >inability to solve Liar's Dice with two dice and four faces each seems a bit of a letdown
>
> **On the contrary.** The message shown is a spurious warning during game tree exploration during setup. **According to the reviewer's comment, the code executes as intended, constructing the game tree, from which the game is solved.** (Perhaps Reviewer dyt9 can concur.) Plus, as explained in our latest reply to Reviewer dyt9, the warning does not impact our calculations. (Edit: **turns out, we misinterpreted the reviewer's comment, but the reviewer did end up running it in their HPC shortly after.** We apologize for the confusion)
>
> >CFR is implemented as an algorithm on the game tree, or on each player's sequence-form polytope
>
> It is implemented as an algorithm on the game tree. We chose to accelerate the naive CFR implementation as done by OpenSpiel and CFR+'s pseudocode [1]. Our understanding is that optimizations explored by Johanson et al. [2] require domain-specific knowledge and compatibility.
>
> [1] O. Tammelin. Solving Large Imperfect Information Games Using CFR+, 2014. URL: https://arxiv.org/pdf/1407.5042
>
> [2] M. Johanson et al. Accelerating Best Response Calculation in Large Extensive Games, 2011. URL: http://johanson.ca/publications/poker/2011-ijcai-abr/2011-ijcai-abr.pdf
>
> >comparing a sophisticated GPU implementation against OpenSpiel's "vanilla" Python implementation
>
> Please note that we also benchmark their C++ implementation which is orders of magnitudes faster than their Python implementation (OpenSpiel offers 2 implementations). Our GPU implementation is shown to perform orders of magnitudes faster than their said C++ implementation as well (up to 114.2 times). (edit: now 203.6 times).
>
> >Certain low-hanging benchmarks have been left out (such as turning on tensor cores, or ablating the choice of floating point format).
>
> CuPy sparse matrices are not compatible with tensor cores, and only 4-byte and 8-byte floating-point numbers are supported. Since our implementation uses 8-byte floats, we are left with 4-byte floats whose results can be newly added to our manuscript.
>
> Again, our rationale is that OpenSpiel uses 8-byte floats, thus making our comparison fairer. But, we are open to adding a new 4-byte float result since this was brought up multiple times. While we will try to incorporate it before the rebuttal revision deadline, we are confident they can be added before the camera-ready. (Edit: **new results with 4-byte floats are added to our latest revision.**)
>
> We hope this clarifies some of the points you raised.

---

> > ### Comment · Reviewer_dyt9 · 2024-11-27
> >
> > > On the contrary. The message shown is a spurious warning during game tree exploration during setup. According to the reviewer's comment, the code executes as intended, constructing the game tree, from which the game is solved. (Perhaps Reviewer dyt9 can concur.) Plus, as explained in our latest reply to Reviewer dyt9, the warning does not impact our calculations.
> >
> > I was only able to do CFR iterations for 2 dice 3 faces (which was slower than the rebel code by an order of magnitude). I was only able to create the `FiniteExtensiveFormGame` which already took too much time. creating the CFR matrices seems to take even more time.

---

> ### Author Response · Authors · 2024-11-27
>
> >slower than the rebel code by an order of magnitude
>
> Again, we struggle to understand why Reviewer dyt9 insists on comparing CFR runtimes with ReBeL runtimes -- a completely different algorithm.
>
> >I tried running the code with liars dice 2 dices and 4 faces but other than warning messages like OpenSpiel exception: /project/open_spiel/games/liars_dice/liars_dice.cc:279 player >= 0, the code would take forever just to build the tree (it took 12 minutes and 70GB of memory, iterating over the game itself only takes 1.5 minutes).
>
> We apologize for the misunderstanding. We thought the above quote indicated you at least ran CFR for a number of iterations.
>
> As we have expressed before, the one-time initial game definition should be considered separate from CFR. Even if it is considered in the runtime, it is more than made up for by the time saved during subsequent CFR iterations (see large IIGs we test in Experiment 2).
>
> Since it was brought up, we will try to add a new result with liar's dice with 2 dice and 4 faces before the rebuttal deadline. (Edit: initial comment omitted 2 dice).
>
> Again, we would like to thank the reviewer for the clarifications.

---

> > ### Comment · Reviewer_dyt9 · 2024-11-27
> >
> > Note that its 2 dice 4 faces,
> >
> > It finally ran but building the cfr matrix took 4700 second, it was able to do ~2.2 seconds per iteration.

---

> > ### Comment · Reviewer_dyt9 · 2024-11-27
> >
> > > Again, we struggle to understand why Reviewer dyt9 insists on comparing CFR runtimes with ReBeL runtimes -- a completely different algorithm.
> >
> > It's the CFR implementation used in rebel (for ld), it's the same algorithm, w/o any truncation.

---

> ### Author Response · Authors · 2024-11-27
>
> Thank you for the clarification! **We're glad that our code for 2 dice 4 faces runs on your HPC, achieving ~2.2 sec/it.** Let's continue our conversation in your comment chain regarding other issues that were brought up.

---

### Meta-Review · Area_Chair_i3Jj · 2024-12-22

**Metareview:**

Before going into details, I want to thank again the reviewers and the authors for engaging so actively on this paper.

My position on this paper was summarized in detail in my post during the response period. To restate the crux of it, from my point of view, the main merit of the authors' contribution is that it would provide the community with an open-source implementation of CFR that is significantly faster than the default Python one in OpenSpiel, while retaining compatibility with OpenSpiel games. Furthermore, I think that the authors have addressed many of the points that the reviewers brought up.

After the end of the response period, I contacted Reviewer dyt9 to try to understand better their position, and we had an in-depth conversation. Overall, I think I made progress in convincing them that the community would find the contribution useful. However, they still seemed not enthusiastic about the paper. From my understanding, their point of view is that nobody that really needs to squeeze performance out of a game solver would use or compare against OpenSpiel's implementations. And on top, they were not in love with the graph-based approach used in this paper to implement CFR, and thought that a level-based approach would do better. Finally, they didn't love the fact that alternation was not supported.

Overall, while I like the paper, for me to override five reviewers and push this paper through seems extreme. The review panel included knowledgeable people on extensive-form games, and I know that Reviewer dyt9 has hands-on experience with implementing GPU-accelerated first-order optimization solvers.

As a concrete suggestion for future iterations, I would recommend implementing alternation, both as a way to dismiss concerns that the approach used in the paper is too restrictive, and also since alternation usually performs well in extensive-form games. It might also help to discuss different design that have been considered and why they were discarded; such a discussion could also help future implementations by other groups as well.

**Additional Comments On Reviewer Discussion:**

The reviewer discussion is summarized in my post below.

---

### Decision · Program_Chairs · 2025-01-22

Reject